# LLM-based Multi-Agent Blackboard System for Information Discovery in Data Science

## Abstract

The rapid advancement of Large Language Models (LLMs) has opened new opportunities in data science, yet their practical deployment is often constrained by the challenge of discovering relevant data within large data lakes. Existing methods struggle with this: single-agent systems are quickly overwhelmed by large, heterogeneous files in the data lakes, while multi-agent systems designed based on a master–slave paradigm depend on a rigid central controller that requires precise knowledge of each sub-agent's capabilities. To address these limitations, we propose a novel multi-agent communication paradigm inspired by the blackboard architecture for traditional AI models. In this framework, a central agent posts requests to a shared blackboard, and autonomous subordinate agents–either responsible for a partition of the data lake or general information retrieval–volunteer to respond based on their capabilities. This design improves scalability and flexibility by eliminating the need for a central coordinator to have prior knowledge of all sub-agents expertise. We evaluate our method on three benchmarks that require explicit data discovery: KramaBench and modified versions of DSBench and DA-Code to incorporate data discovery. Experimental results demonstrate that the blackboard architecture substantially outperforms baselines, including RAG and the master–slave multi-agent paradigm, achieving between 13% to 57% relative improvement in end-to-end task success and up to a 9% relative gain in F1 score for data discovery over the best-performing baselines across both proprietary and open-source LLMs. Our findings establish the blackboard paradigm as a scalable and generalizable communication framework for multi-agent systems.

## 1 Introduction

The recent developments in Large Language Models (LLMs) have introduced new paradigms for data science workflows, enabling natural language-based approaches to data interpretation, transformation, and analysis (Jing et al., 2025; Huang et al., 2024; Hong et al., 2025; Wang et al., 2025). Existing work, however, typically assumes an idealized setting in which relevant datasets are already curated and provided to the model—an assumption that diverges substantially from the practical challenges encountered in real-world data science (Lai et al., 2025). In practice, a substantial fraction of effort is devoted to locating the appropriate data within large and heterogeneous data lakes, often comprising thousands of loosely organized files—a process that constitutes a major bottleneck before any downstream analysis can be performed (Xu et al., 2021). We argue that this stage of data discovery is both a critical and underexplored challenge for applying LLMs effectively.

Previous work on data science tasks that require discovery[1] from a data lake has primarily relied on single-agent systems in which an LLM is given access to all candidate files within its context window and is then asked to solve the problem (Lai et al., 2025). This method suffers from several limitations. First, it is not scalable: as the number of files grows, fitting them into the limited context window of an LLM becomes infeasible. Second, the heterogeneity of files poses a challenge, as a single agent may struggle to effectively analyze, interpret, and integrate diverse forms of information. Third, such systems lack robustness to noise, since the presence of many irrelevant files can overwhelm the model and degrade both reasoning quality and precision. One may argue that Retrieval-Augmented Generation (RAG) (Lewis et al., 2020; Kim et al., 2024; Salemi & Zamani, 2024b) provides a solution by choosing a subset of files in the data lake; However, current retrieval techniques are known to perform poorly on tabular and domain-specific data, which are pervasive in data science applications (Yu et al., 2025; Ji et al., 2025; Huang et al., 2022; Gu et al., 2025).

---

[1]Some example tasks are shown in Figure 13 and 14 in Appendix D. They require computing or aggregating information from raw data within a large data lake, where the specific source files are not pre-identified.

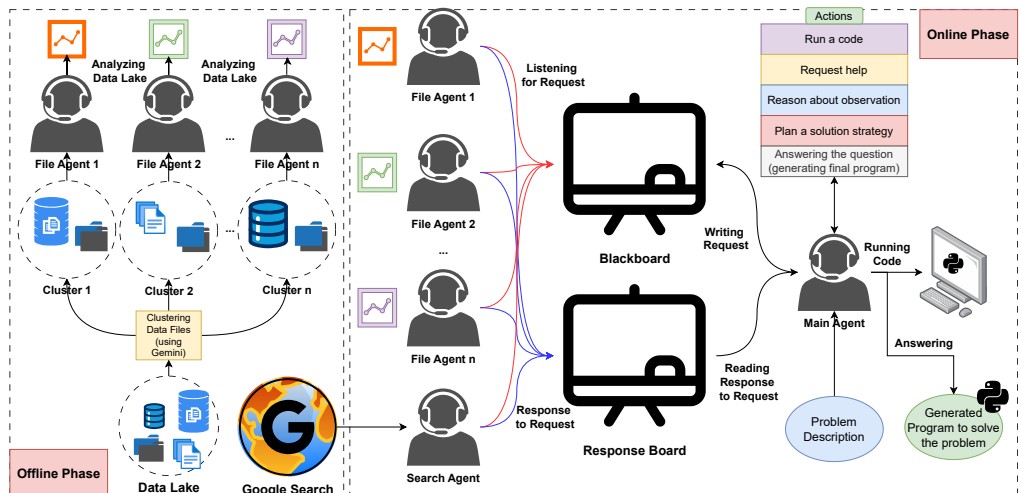

Figure 1: Overview of the blackboard multi-agent system for information discovery in data science. In this framework, the main agent does not assign tasks to subordinate agents. Instead, it posts requests to the blackboard, and subordinate agents autonomously decide whether to respond based on their expertise. The main agent then uses the responses to the request to solve the given task.

An alternative approach explores multi-agent systems, which frequently adopt a master–slave paradigm (Li et al., 2024; Han et al., 2025; Xu & Peng, 2025). In this setting, a single controller (e.g., orchestration agent) assigns subtasks to a set of subordinate agents that then execute the specified actions. While conceptually straightforward, this architecture has several drawbacks. First, this master-slave paradigm limits the agents' autonomy: subordinate agents are forced to execute instructions from the coordinator even when they lack sufficient information or hold outdated or erroneous information. Second, the central controller must maintain an accurate model of each agents capabilities to assign tasks, an assumption that is often unrealistic when agents have only partial or evolving knowledge of the problem space. Finally, when multiple agents possess overlapping expertise, the controller faces an inherent assignment ambiguity, making task routing difficult.

Inspired by the blackboard architecture that with substantial impact on traditional AI systems since the 1980s (Erman et al., 1980; Botti et al., 1995), we adopt a new communication paradigm for LLM multi-agent systems. In this paradigm, a central agent remains responsible for solving the overall task, similar to the master-slave paradigm. However, rather than assigning subtasks to specific agents, the central agent posts a request on a shared *blackboard* that describes the task or information needed, as shown in Figure 1. Subordinate agents monitoring the blackboard can independently decide whether they possess the capability, knowledge, or interest to contribute to solving the task. This design shifts decision-making from a single coordinator to a distributed model whose agents autonomously determine their participation, enabling more flexible collaborations. This differs from the conventional shared-memory paradigm in multi-agent systems. In shared-memory (Sagirova et al., 2025), agents perform assigned tasks based on information in the shared memory, effectively being asked to execute tasks determined by a central coordinator. Conversely, *in the blackboard architecture, there is no task assignment; instead, requests are broadcast on the blackboard, and each agent retains full autonomy to decide whether to participate in solving the task or not.*

While the blackboard architecture can be applied broadly within multi-agent frameworks, its application to data science with data discovery is particularly compelling and underexplored. As shown in Figure 1, the data lake can be partitioned into smaller clusters, e.g., based on similarity, homogeneity, or any criteria that facilitate efficient handling, each assigned to a subordinate agent responsible for understanding and processing that subset. The main agent, which is tasked with solving the given problem, posts requests on the blackboard specifying the data or general information required. Subordinate agents with the relevant knowledge or capability then autonomously volunteer to respond. This design ensures that each sub-agent manages only a subset of files or web-based information, enhancing scalability compared to approaches that require all data to be loaded into the main agent's prompt. Importantly, the main agent does not need prior knowledge of sub agents knowledge or capability to solve the task, simplifying coordination and improving flexibility in large-scale data

lake environments. Here, the main agent's role is primarily to describe the information it requires and define tasks for the sub agents, without directly managing or assigning tasks to them.

We conduct experiments on three datasets for data science tasks that require an explicit information discovery phase. KramaBench (Lai et al., 2025) is a recently released benchmark designed for this purpose and, to the best of our knowledge, the only publicly available dataset that directly evaluates data discovery in data science. In addition, we repurpose DSBench (Jing et al., 2025) and DA-Code (Huang et al., 2024) by introducing a data discovery component, thereby making them more challenging than their original formulations. Experimental results across these datasets demonstrate that the proposed blackboard architecture consistently outperforms strong baselines, including RAG and the master-slave multi-agent framework, achieving 13% to 57% relative improvement over the best performing baseline in end-to-end problem solving depending on the backbone LLM. Notably, this improvement is observed across both proprietary and open-source LLMs, highlighting the generalizability of the approach. Furthermore, our method also surpasses baselines in data discovery performance, yielding up to a 9% relative gain in F1 score for correctly identifying relevant files from the data lake. These results underscore the effectiveness of the blackboard architecture as a communication paradigm for multi-agent systems in data science.

## 2 PROBLEM FORMULATION

Let $\mathbb{D} = \{d_i\}_{i=1}^N$ denote a data lake consisting of $N$ distinct data files, each containing information potentially completely or partially relevant to answering a data science question $q$ (some examples of these questions are shown in Figures 13 and 14 in Appendix D). The objective of this work is to design a generative system $\pi_s$ that, given the query $q$ and the data lake $\mathbb{D}$ as input, produces a program $p \sim \pi_s(q; \mathbb{D})$ in response. When executed (e.g., using a Python interpreter in this paper), this program $p$ retrieves, loads, and processes the appropriate data from the data lake $\mathbb{D}$ and solve the given problem in the question $q$ to compute the answer. To evaluate the generated program $p$, we assume the existence of an evaluation function $\mu_{\text{generation}}$ that executes $p$ to produce an output $o_p$, compares $o_p$ with the ground-truth response $y_q$, and assigns a corresponding score. In addition, we assume a metric $\mu_{\text{retrieval}}$, given the program $p$ and the ground-truth files $\mathbb{D}_q$, assigns a score reflecting the performance in discovering the correct data sources.

## 3 LLM-BASED MULTI-AGENT BLACKBOARD SYSTEM

This section introduces an alternative communication paradigm for LLM-based multi-agent systems inspired by blackboard systems (Erman et al., 1980), distinct from the widely used master–slave architecture. As outlined in §1, blackboard-based multi-agent systems provide several advantages over the master-slave approach. Here, rather than directly assigning tasks to sub-agents, the main agent posts its requests (i.e., sub-tasks for which it requires assistance) on a shared blackboard, which functions as a broadcast channel accessible to all other agents. Each helper agent independently evaluates whether it can respond to a request, considering its own capabilities, availability, cost, and other factors. If an agent decides to contribute, it writes its response to the corresponding request, and the main agent then decides whether to use or ignore the provided information. *This way, all agents in the system retain full autonomy over their actions, and no centralized controller forces them to execute a specific task.* While the blackboard paradigm is applicable to a wide range of multi-agent systems, we focus on data science tasks that require data discovery, where its characteristics are particularly advantageous, as discussed in §1. The remainder of this section details our method and its design for data science problems that require information discovery.

**Overview:** An overview of our proposed method is presented in Figure 1. The system $\pi_s$ operates over the data lake $\mathbb{D}$ by first partitioning $\mathbb{D}$ into $C$ clusters of related files. Each cluster $\mathbb{D}_i$ is assigned to a file agent $\pi_{f_i}$, which is responsible for handling, loading, processing, and retrieving information from the files within its cluster. In addition, a search agent $\pi_s$ is included to retrieve external information from the web that may be required to solve the problem. The overall system $\pi_s$ is composed of a main agent $\pi_m$, which is responsible for solving the query $q$, and a set of $C + 1$ helper agents $\Pi_{\text{helper}} = \{\pi_{f_i}\}_{i=1}^M \cup \{\pi_s\}$ that provide specialized assistance. The query $q$ is presented to $\pi_m$, which iteratively selects an action $a \in \mathbb{A}$ from the action space $\mathbb{A}$, executes it, and observes the resulting outcome. Among its actions, the main agent may interact with a

blackboard $\beta$, a shared communication medium where it can post a request $r$ without addressing a specific sub-agent. The helper agents $\Pi_{\text{helper}}$ continuously monitor the blackboard, determine whether they can address a posted request, and, if so, provide their outputs on the corresponding response board $\beta_r$. These responses are then collected and made available to $\pi_m$, which incorporates them into its decision-making process.[2] The main agent is limited to at most $T$ sequential actions (including blackboard interactions) to solve the query $q$, ultimately producing a program $p$ in python programming language that computes the final answer to $q$.

**Clustering Data Lake:** There are multiple approaches for partitioning the data lake into clusters; applying clustering algorithms over file representations, random partitioning, or other heuristic methods. For simplicity, we do not utilize file content and instead rely solely on file names. Specifically, the file names are provided to an LLM—Gemini-2.5-Pro[3]—which using the prompt shown in Figure 5, clusters the files into categories based only on their names.[4] An example of this clustering is provided in Figure 12 in Appendix D, where the model successfully groups related files together. For instance, it clusters all files originating from the National Interagency Fire Center into a category labeled "NIFC Wildfire Statistics." The number of automatically derived clusters for each dataset is reported in Table 3 in Appendix A.

### 3.1 Main Agent

The primary role of the main agent is to solve the problem in collaboration with the helper agents. The main agent follows the ReAct framework (Yao et al., 2023), where at each step $t$, given the query $q$ and the history of actions and observations $\mathbb{H}_{t-1}$, it first reasons about what is the best next action and selects an action from a predefined action space, executes the action, observes the outcome, and appends the resulting observation to update the history $\mathbb{H}_t$.[5] The prompt used by the main agent is shown in Figure 6 in Appendix B. The agent selects one of the following predefined actions in each step, executes them, and observe their outcomes:

- *Planning:* In this action, the LLM decomposes the problem into smaller sub-problems and outlines a plan for addressing each of them. This action has no external effect on the environment but serves as an internal reasoning step to guide the LLM's problem-solving process. In response, the system simply acknowledges the proposed plan and instructs the LLM to proceed.

- *Reasoning:* In this action, the LLM focuses on a specific aspect of the problem and explains its reasoning, analysis, or interpretation of the available observations and steps taken so far in this process. Similar to the planning step, this action has no external effect on the environment but functions as an internal reasoning mechanism to guide the LLM's problem-solving process. In response, the system simply acknowledges the reasoning and prompts the LLM to continue.

- *Executing Code:* In this action, the agent generates python code, which is executed using a python interpreter. If the code runs successfully, the resulting outputs are returned to the agent for observation; otherwise, the agent receives the corresponding error messages. This action enables the agent to explore the problem interactively, inspect data files, and experiment with them to gain a deeper understanding of their content and structure and how to process them.

- *Requesting Help:* In this action, the agent formulates a request for assistance from the sub-agents, specifying, for example, the types of data files or information needed, or the resources required to apply a tool or solve a sub-problem. This request is posted on the blackboard $\beta$ for visibility by the helper agents. Once the sub-agents respond, if they respond, their responses on the response board $\beta_r$ are collected and provided back to the main agent as the outcome of this action for observation and further use in its decision-making process.

---

[2]Responses are not written back to the blackboard $\beta$ to avoid dependencies where one sub-agent's output could influence the behavior of others negatively. Instead, all responses are directed exclusively to the response board $\beta_r$, ensuring independent operation of sub-agents and exclusive access by the main agent $\pi_m$.

[3]Available at: https://cloud.google.com/vertex-ai/generative-ai/docs/models/gemini/2-5-pro

[4]This method represents just one possible approach to clustering, chosen for simplicity; more scalable alternatives could equally be employed in real world scenarios.

[5]In this work, the inputs, outputs of the model, and observations are appended directly to the prompt of the LLM, formatted according to its chat-based input template.

- **Answering:** In this action, the agent concludes the problem-solving process by generating a final program that produces the answer to the query. This action terminates the process, and the output of this step constitutes the final program $p$ generated by the system to address the problem.

## 3.2 HELPER AGENTS

In a data science, information discovery can typically be categorized into two tasks: (1) identifying the specific files that contain the data necessary to the problem, and (2) retrieving general knowledge about concepts relevant to the problem, such as domain-specific terms or details of particular algorithms and methods. To support these, our framework employs two types of helper agents:

**File Agent:** Handling all the files in a data lake with a single agent is not feasible for several reasons: it typically involve a large number of files, many of which are lengthy and may exceed the agents context window; the files span diverse topics, which can confuse the agent and hinder effective reasoning; and accessing and processing all files simultaneously can be computationally expensive and inefficient, leading to unnecessary overhead and slower problem-solving. For these reasons, in our framework each file agent is assigned responsibility for a subset of data files determined to be relevant, as described earlier in the clustering procedure. In an offline phase, the file agent $\pi_{f_i}$ takes as input a subset of the data lake $\mathbb{D}_i$ and operates through a two-step procedure. In the first step, the agent selects a subset[6] (or all) of the files to examine their content. The contents of them are presented to the agent for inspection (details of presentation are in Appendix C). In the second step, after observing the selected files, the agent reasons about and analyzes them, learning how they are structured, what pre-processing or transformations may be required, and how they should be processed in general. An example of such an analysis is provided in Figure 11 in Appendix D. Then, in the online phase, the agent listens for requests from the main agent. Upon receiving a request, based on the analysis it did earlier, it determines whether it can contribute to answering it. If so, the agent generates a detailed plan specifying which files in $\mathbb{D}_i$ are relevant, how they should be loaded in Python code, what libraries to use, the steps required for data processing, and samples from the data. The prompt used to guide the file agent is shown in Figure 7 in Appendix B.

**Search Agent:** Certain data science problems require task-specific knowledge about algorithms or domain expertise that the LLM may not possess. To address this, we design a web-search agent that retrieves relevant information from a search engine. This agent operates according to the prompt shown in Figure 8 in Appendix B. Given a request $r$ posted on the blackboard $\beta$, the agent first determines whether it is capable of addressing the request. It is specifically restricted to general web-based information retrieval and does not respond to requests involving access to local files or datasets. If the agent determines that the request can be answered, it enters an iterative search process with a maximum of $T_{\text{search}} = 3$ steps. At each step $t$, the agent generates a set of queries $\mathbb{Q}_t$, which are submitted to a search engine—in this work, Google Custom Search Engine[7]—to retrieve $k = 3$ webpage per query. The content of the webpages are then extracted using *beautifulsoup* library[8] to be presented to the search agent. The extracted documents are then evaluated by the agent to determine whether they provide sufficient information to answer the request. If so, the agent generates a response to the request, which is posted to the response board $\beta_r$. If the information is insufficient, a new set of queries is generated to continue gathering relevant data from the web.

## 4 EXPERIMENTS

### 4.1 EXPERIMENTAL SETUP

**Benchmarks:** To the best of our knowledge, KramaBench is the only public benchmark for data science problems that explicitly incorporates a data discovery phase, which we adopt in our evaluation. In addition, we repurpose two existing datasets, DSBench (data analysis task) (Jing et al., 2025)

---

[6]When filenames indicate multiple files containing the same type of data over different time periods, the agent does not need to inspect all of them to infer the structure; a small representative sample is sufficient.

[7]We use Google Custom Search Engine, configured to exclude all websites associated with the datasets used in this paper to prevent data leakage: https://developers.google.com/custom-search

[8]Available at: https://pypi.org/project/beautifulsoup4/

and DA-Code (Huang et al., 2024), to include in this phase. Specifically, we manually filtered out all questions that do not require any data file for answering, as well as those that lack sufficient hints for data discovery.[9] After filtering, we aggregated all remaining files across questions into a unified data lake, such that the model must perform discovery to identify relevant files at inference time. In this setup, only the question and the data lake are provided to the model, requiring it to identify the relevant files to answer the question, following the same protocol as KramaBench. Further details on this filtering process, along with dataset statistics in Table 3, are provided in Appendix A.

**Evaluation:** To evaluate the generated programs, we execute each and compare its output against the ground-truth reference for the corresponding question. For each dataset, we adopt its standard evaluation protocol. For KramaBench, we use the official evaluation script provided in its repository.[10] For DA-Code, we likewise rely on the official evaluation script released by its authors.[11] For DSBench, we use the original evaluation method, in which an LLM serves as the judge. The generated programs output is compared against the reference answer using Gemini-2.5-Pro as the judge LLM, with the evaluation prompt shown in Figure 4 in Appendix A, producing a binary score.

**Inference Setup:** We set the maximum actions of the main agent to $T = 10$. We use nucleus sampling (Holtzman et al., 2020) with a temperature of 0.1 for more deterministic inference and default value for other hyperparameters. Proprietary models are accessed via Vertex AI,[12] while open-source models are served by vLLM.[13] At each step, we cap the number of generated tokens at 8,192. We use Gemini-2.5-Pro and -Flash (Gemini-Team, 2025), and Claude-4-Opus (Anthropic, 2025) as the proprietary and Qwen3-Coder[14] with 30 billion parameters (Qwen-Team, 2025) as the open-source LLMs. Experiments are conducted on 2 NVIDIA A100 (80GB VRAM) GPUs.

**Baselines:** To evaluate our method against alternative approaches for solving data science problems involving data discovery, we compare it with the following baselines:

- **DS-GRU:** We adopt the only existing baseline (to the best of our knowledge) for data discovery in data science problems, which appends all available files directly into the LLM prompt and attempts to solve the problem (Lai et al., 2025). This baseline uses a self-correction loop that retries when errors occur in generated codes. For details, we refer the reader to Lai et al. (2025).

- **Retrieval-Augmented Generation (RAG):** This retrieves the top 5 files[15] based on the file names and contents (the method for presenting a file content to the LLM is explained in Appendix C) from the data lake using E5-large[16] (Wang et al., 2022), a 330M-parameter embedding model and use it to solve the problem. It then follows the same procedure as the main agent described in Section 3.1, with two key modification: 1) the retrieved files contents and addresses are presented directly to the LLM in the prompt and 2) the general help-request action is replaced with a restricted action that only allows direct requests to the search agent. This design isolates the effect of substituting the file discovery mechanism with RAG, enabling a controlled study of its impact on performance. The prompt used for this baseline is shown in Figure 10 in Appendix B.

- **Master-Slave:** This baseline follows the same procedure as the main agent described in Section 3.1. The key difference is that, instead of posting requests on the blackboard, the agent directly invokes sub-agents (consisting of the search agent and the file agents as explained in Section 3.2) based on their description by referencing their names and assign task to them. The prompt used for this baseline is shown in Figure 9 in Appendix B.

---

[9]For example, questions that request the computation of a data science metric on a column without specifying the structure or content of the relevant file.

[10]Available at: https://github.com/mitdbg/KramaBench

[11]Available at: https://github.com/yiyihum/da-code

[12]Available at: https://cloud.google.com/vertex-ai?hl=en

[13]Available at: https://docs.vllm.ai/en/latest/

[14]Available at: https://huggingface.co/Qwen/Qwen3-Coder-30B-A3B-Instruct

[15]This number is chosen based on the average number of files required to solve the problems (1.6) and the length of the context window of the backbone LLMs used in this paper.

[16]Available at: https://huggingface.co/intfloat/e5-large-v2

Table 1: Results on the KramaBench, DSBench, and DA-Code benchmarks. The best results for each LLM are highlighted in **bold**. The KramaBench categories are abbreviated: Arc. (Archaeology), Ast. (Astronomy), Bio. (Biomedical), Env. (Environment), Leg. (Legal), and Wild. (Wildfire).

| | Method | LLM | KramaBench | | | | | | | DS Bench | DA-Code | Average (macro) |
| | | | Arc. | Ast. | Bio. | Env. | Leg. | Wild. | Average | | | |
|---|---|---|---|---|---|---|---|---|---|---|---|---|
| (1) | DS-GRU | | **0.00%** | 1.80% | 2.11% | 1.15% | 3.27% | 13.54% | 3.64% | 0.00% | 0.00% | 1.21% |
| (2) | RAG | Qwen3- | **0.00%** | 3.16% | 4.99% | 0.54% | 6.19% | 16.93% | 5.30% | 6.32% | 0.00% | 3.87% |
| (3) | Master-Slave | Coder | **0.00%** | 3.55% | 3.39% | **7.77%** | **8.90%** | 21.79% | 7.56% | 7.55% | 0.00% | 5.03% |
| (4) | Blackboard | | **0.00%** | **7.69%** | **7.85%** | 4.47% | 6.36% | **23.97%** | **8.39%** | **14.22%** | **1.11%** | **7.90%** |
| (5) | DS-GRU | | 0.00% | **7.83%** | 0.09% | 10.93% | 12.46% | 13.34% | 7.44% | 5.53% | 0.00% | 4.32% |
| (6) | RAG | Gemini 2.5 | 16.66% | 3.57% | 13.98% | **28.57%** | 10.97% | 33.67% | 17.90% | 22.92% | **2.75%** | 14.52% |
| (7) | Master-Slave | Flash | 16.66% | 3.16% | 13.98% | 17.46% | 21.75% | 25.80% | 16.46% | 26.48% | 0.55% | 14.49% |
| (8) | Blackboard | | 16.66% | 3.57% | **14.78%** | 22.92% | **27.09%** | **41.04%** | **21.01%** | **28.06%** | 0.55% | **16.54%** |
| (9) | DS-GRU | | 25.00% | 6.69% | 10.64% | 27.47% | 5.94% | 39.36% | 19.18% | 3.95% | 0.00% | 7.71% |
| (10) | RAG | | 33.33% | 8.47% | 32.53% | 31.36% | 25.55% | 38.32% | 28.26% | 27.27% | 0.00% | 18.51% |
| (11) | Master-Slave | Gemini 2.5 | 33.33% | 8.47% | 24.74% | 32.81% | 34.64% | 58.98% | 32.16% | 34.38% | 5.49% | 24.01% |
| (12) | Blackboard | Pro | 33.33% | **17.95%** | **36.83%** | **39.31%** | 34.92% | **62.88%** | **37.53%** | **38.73%** | **9.34%** | **28.53%** |
| (13) | DS-GRU | | 8.33% | 1.38% | 1.90% | 8.14% | 9.80% | 23.14% | 8.78% | 3.55% | 0.00% | 4.11% |
| (14) | RAG | Claude 4 | 33.33% | 11.52% | 23.42% | 31.61% | 31.80% | 45.80% | 29.58% | 35.57% | 3.85% | 23.00% |
| (15) | Master-Slave | Opus | 33.33% | 8.69% | 32.28% | **39.16%** | **44.08%** | 48.35% | 34.31% | 45.84% | 2.75% | 27.63% |
| (16) | Blackboard | | 33.33% | **18.69%** | **45.31%** | 34.35% | 42.48% | **50.06%** | **37.37%** | **49.80%** | 7.14% | **31.43%** |

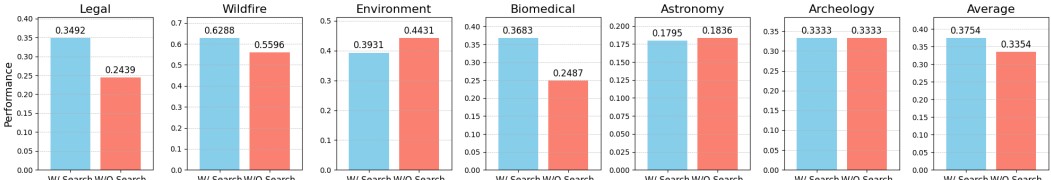

Figure 2: Performance of Blackboard System w/ and w/o search agent (Gemini 2.5 Pro).

## 4.2 EMPIRICAL FINDINGS

**Main Results:** We conduct our experiments on the datasets described in Section 4.1 using our method and the baselines. The results are presented in Table 1. These results demonstrate that our method, the Blackboard System, outperforms all baselines on average across all the datasets. Specifically, the Blackboard System surpasses the DS-GRU, RAG and Master-Slave approaches on all three datasets and achieves similar or higher performance in 4 out of 6 categories on KramaBench. Furthermore, we observe that the Blackboard System consistently outperforms the baselines regardless of the backbone LLM, highlighting its robustness and generalizability. We attribute this improvement to the design of the Blackboard System, where tasks are not explicitly assigned to helper agents; instead, each agent autonomously decides whether to participate based on its capabilities. This self-selection enhances both problem-solving efficiency and data discovery performance.

**File Discovery Performance:** To analyze the effectiveness of different methods in data discovery, we report recall, precision, and F1-score for the file discovery task, i.e., identifying the correct files required to answer each question. The results of this experiment, using Gemini 2.5 Pro as the backbone LLM, are presented in Table 2. The results in this table indicate that the blackboard system achieves the highest recall, precision, and F1-score compared to all baselines, both on average and across the three datasets. In particular, for KramaBench, the blackboard system attains the highest F1-score in 4 out of 6 domains. We attribute this improvement to the design of the blackboard system, where the main agent does not directly assign requests to specific file agents, as in the master–slave setup. Instead, each file agent independently decides whether it can contribute based on its capabilities and data holdings, leading to more accurate and comprehensive file discovery.

**Effect of Web Search (Search Agent) on the Performance:** We observed that in some cases the backbone LLM lacks the necessary domain-specific knowledge or familiarity with specialized algorithms to fully understand and solve the problem. To address this limitation, the inclusion of a search agent that can retrieve relevant external information may be beneficial. To evaluate this, we compare the blackboard system with and without the search agent. The results on KramaBench, shown in Figure 2 using Gemini 2.5 Pro as the backbone LLM, demonstrate that incorporating the search agent improves the average performance of the blackboard system. Further analysis

Table 2: File discovery performance, obtained using Gemini-2.5-Pro as the LLM. The best results are highlighted in **bold**. The KramaBench categories are abbreviated: Arc. (Archaeology), Ast. (Astronomy), Bio. (Biomedical), Env. (Environment), Leg. (Legal), and Wild. (Wildfire).

| | Method | Metric | KramaBench | | | | | | | DS-Bench | DA-Code | Average (macro) |
| | | | Arc. | Ast. | Bio. | Env. | Leg. | Wild. | Average | | | |
|---|---|---|---|---|---|---|---|---|---|---|---|---|
| (1) | RAG | recall | 0.875 | 0.125 | **0.666** | 0.3506 | 0.127 | 0.238 | 0.396 | 0.035 | 0.257 | 0.229 |
| | | precision | **1.000** | 0.125 | 0.666 | 0.450 | 0.133 | 0.452 | 0.471 | 0.047 | 0.456 | 0.324 |
| | | F1 | 0.916 | 0.125 | 0.629 | 0.332 | 0.105 | 0.301 | 0.401 | 0.034 | 0.307 | 0.247 |
| (2) | Master-Slave | recall | **0.916** | 0.5138 | 0.648 | 0.382 | **0.444** | **0.567** | 0.578 | 0.323 | 0.546 | 0.482 |
| | | precision | 0.930 | **0.750** | **0.722** | 0.500 | **0.494** | **0.642** | 0.673 | 0.503 | 0.767 | 0.647 |
| | | F1 | 0.913 | 0.577 | **0.674** | 0.389 | **0.450** | **0.576** | 0.596 | 0.358 | 0.584 | 0.513 |
| (3) | Black-board | recall | **0.916** | 0.576 | 0.648 | **0.604** | 0.383 | 0.464 | **0.598** | **0.402** | **0.600** | **0.533** |
| | | precision | **1.000** | 0.733 | 0.722 | **0.703** | 0.302 | 0.603 | **0.677** | **0.584** | **0.837** | **0.699** |
| | | F1 | **0.944** | **0.618** | 0.674 | **0.588** | 0.304 | 0.495 | **0.603** | **0.438** | **0.643** | **0.561** |

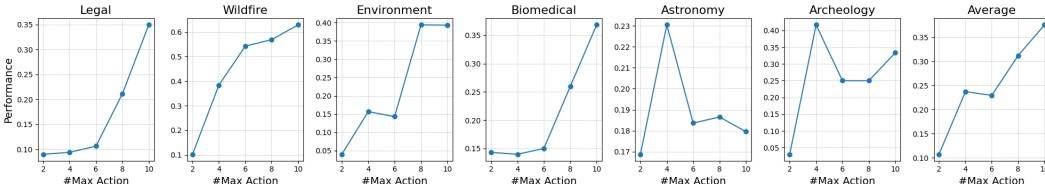

Figure 3: Performance of Blackboard System with various maximum actions by the main agent.

reveals that when the main agent encounters unfamiliar concepts, it issues requests to obtain such information from the web. In these cases, the search agent typically responds by retrieving the required knowledge, thereby enabling the main agent to continue solving the problem effectively. Illustrative examples of this behavior are provided in Figures 13 and 14 in Appendix D, highlighting the importance of the search agent in scenarios where external domain knowledge is essential.

**Effect of Number of Main Agent's Actions on the Performance:** To examine the impact of the maximum number of actions available to the main agent, we vary this parameter across $2, 4, 6, 8, 10$ and evaluate the blackboard system on KramaBench using Gemini 2.5 Pro as the backbone LLM. The results, presented in Figure 3, indicate that increasing the action budget consistently improves the average performance of the system. This trend aligns with intuition: a larger exploration budget allows the agent to more thoroughly analyze the problem, consider alternative strategies, better investigate the solution space, and generate a better program that answers the question.

**Scalability Comparison between Blackboard and Master–Slave Systems:** We report relative (percentage) performance gains rather than absolute score differences to enable a fair comparison across datasets with varying difficulty and score ranges. Absolute improvements are not directly comparable across tasks because they depend on baseline accuracy and the attainable upper bound; for instance, a 5-point increase is minor when the baseline is already high but more meaningful for a challenging task with low initial performance. In contrast, relative gain measures the proportional improvement with respect to the baseline and therefore provides a normalized metric across heterogeneous tasks. Using this normalized measure, we analyze how the Blackboard architecture scales relative to the Master–Slave baseline as a function of data lake size (Figure 19 in Appendix E). Each point in the figure corresponds to a task domain, and the regression line shows a positive correlation between data lake size and relative performance gain. Consistent with this trend, Blackboard achieves its largest gains on the two datasets with the largest data lakes, DA-Code (145 files) and Astronomy from KramaBench (1556 files), with relative improvements of 70% and 211%, respectively. These results indicate that Blackboard scales more effectively in large data lakes, whereas the Master–Slave architecture yields limited additional gains as the data lake grows.

**Runtime and Cost Analysis:** To characterize the efficiency–cost trade-off of the Blackboard system relative to the RAG and Master–Slave baselines, we randomly sample 50 questions from the KramaBench benchmark spanning all 6 different domains in the benchmark and measure both runtime and cost per question for each method. Unlike RAG and Master–Slave, which execute their component calls sequentially following the ReAct framework, the Blackboard architecture parallelizes sub-agent interactions: once the main agent posts a request to the shared blackboard, the

corresponding sub-agents process it independently. As shown in Figure 20 in Appendix E, the runtime of all three systems lies in a narrow range (132.0–145.2 seconds), indicating no substantial difference in latency. In terms of monetary cost, Blackboard is more expensive per question (approximately $2.3\times$ the cost of RAG and $1.8\times$ that of Master–Slave), due to increased token usage. However, this additional cost yields substantial performance gains—54.1% over RAG and 18.8% over Master–Slave—so that Blackboard achieves higher answer quality while maintaining comparable runtime, resulting in a favorable accuracy–cost trade-off.

**Comparison with Advanced Data-Science Baselines:** We further compare the Blackboard system with two advanced multi-agent frameworks for data analysis and reasoning, Data Interpreter (Hong et al., 2025) and AutoGen (Wu et al., 2023). Table 4 in Appendix E reports results on the KramaBench benchmark. The Blackboard system achieves higher performance on five out of six tasks as well as on the overall average, indicating more effective adaptation and coordination across heterogeneous data-science workloads compared to these advanced and competitive baselines.

**Effect of Clustering Strategy and Number of Clusters on Performance:** As described in Section 3, we use Gemini 2.5 Pro to cluster files based on their filenames, since providing full file contents to the model is not feasible. To study content-based clustering, we instead encode each file's content with the E5-Large embedding model (Wang et al., 2022) and apply the KMeans algorithm (Lloyd, 1982) to partition files into a fixed number of clusters. Unlike Gemini, which infers the number of clusters automatically, KMeans requires this value as input; we therefore consider 2, 4, and 8 clusters and restrict our analysis to datasets with at least 100 files (the Legal and Astronomy subsets from KramaBench and DA-Code). Figure 21 in Appendix E shows that increasing the number of clusters generally improves performance, as each sub-agent operates on a smaller, more coherent subset of files. Figure 22 in Appendix E compares filename-based clustering with Gemini against content-based clustering, and shows that semantic content clustering consistently outperforms filename-only clustering, indicating that the framework extends naturally to embedding-based clustering and is not restricted to LLMs for grouping data files.

**Case Studies:** To qualitatively analyze the blackboard system—specifically how it formulates requests and how this process improves the generated program—we present several case studies:

- **Writing Request on the blackboard:** An example of a request posted by on the blackboard is shown in Figure 15 in Appendix D. In this case, the main agent, given the data science question, formulates a request that specifies the likely column names and data formats needed to solve the problem, along with some guidance for interpretation. In response, several helper agents (3 out of 8 in this example) chose to contribute. Although the relevant files were distributed across different clusters managed by different file agents, each responding agent independently provided the file addresses, code snippets for loading the data, and explanations of the data structure along with suggested preprocessing steps. Collectively, these responses covered all the ground-truth files required to answer the question. This case study demonstrates how the main agent can effectively leverage the blackboard mechanism to discover and integrate necessary information.

- **Comparing Generated Program by Blackboard System with Master-Slave System:** To study this further, we present an example of programs generated by the Blackboard system and the Master–Slave system in Figure 16 in Appendix D. In this case, the Blackboard agent achieved a better solution because it accurately interpreted the prompt and selected the correct data files. Specifically, it identified that the patients `Age` was located in the `mmc1.xlsx` file and, more importantly, that the requested `APP-Z` score was in the `mmc7.xlsx` file. In contrast, the Master–Slave agent misinterpreted the request and instead used a general protein abundance score (`APP_log2_abundance`) from the wrong file, `mmc2.xlsx`. This critical error in data selection led the Master–Slave agent to produce an incorrect result of `74`, while the Blackboard agents precise data discovery and reasoning yielded the correct answer of `60`.

## 5 RELATED WORK

**LLMs for Data Science:** Specialized benchmarks have emerged to evaluate LLMs in data science. DS-1000 (Lai et al., 2023), ARCADE (Yin et al., 2023), DataSciBench (Zhang et al., 2025), and DSEval (Zhang et al., 2024) assess the translation of natural language instructions into correct

implementations, distinguishing them from broader programming benchmarks such as SWE-Bench (Jimenez et al., 2024), ML-Bench (Tang et al., 2025), and BigCodeBench (Zhuo et al., 2025). While most assume that the relevant data files are pre-specified, recent efforts address multi-step reasoning: DSBench (Jing et al., 2025) and BLADE (Gu et al., 2024) evaluate implementation planning, and ScienceAgentBench (Chen et al., 2025) and BixBench (Mitchener et al., 2025) focus on integrating domain knowledge. These benchmarks, however, still overlook the practical challenge of discovering relevant data within large, heterogeneous repositories—a gap addressed by KramaBench (Lai et al., 2025), which explicitly evaluates data discovery. Building on this, we study how agents can autonomously identify and leverage the correct data sources for end-to-end analysis.

Applications of LLMs in data science have evolved from single-turn code generation to interactive, tool-augmented agents that exploit models specialized for code, including GPT (Brown et al., 2020), CodeGen (Rubavicius et al., 2025), StarCoder (Li et al., 2023), and Code Llama (Rozière et al., 2024). While few-shot prompting (Brown et al., 2020) remains effective, state-of-the-art approaches adopt agentic or multi-agentic frameworks that combine iterative reasoning with external tool use. ReAct (Yao et al., 2023) pioneered the interleaving of reasoning and action, later extended to execution environments (Chen et al., 2019). Toolformer (Schick et al., 2023) and Gorilla (Patil et al., 2024) explicitly train LLMs to call APIs, a capability critical for tasks relying on specialized libraries. Self-correction is a another key feature: frameworks like Self-Debug (Chen et al., 2024) and Reflexion (Shinn et al., 2023) refine generated code using execution feedback. To further enhance reliability, many systems integrate RAG (Lewis et al., 2020; Salemi et al., 2025; Salemi & Zamani, 2025; 2024a) to retrieve documentation or code examples, reducing hallucinations and ensuring up-to-date library use. Additionally, multi-agent master-slave frameworks, such as AutoKaggle (Li et al., 2024), have demonstrated promising results in addressing these challenges.

**Blackboard Systems:**  The blackboard system is a seminal architectural model from classical AI, developed for complex problems that require incremental and opportunistic reasoning. It was implemented in the Hearsay-II speech understanding (Erman et al., 1980) and is characterized by three components: (1) a global, hierarchical data structure (the blackboard) that maintains the current state of the solution; (2) independent specialist modules, known as knowledge sources, which monitor the blackboard and contribute partial solutions; and (3) a control mechanism that opportunistically determines which knowledge source to activate next (Nii, 1986). Following successful applications in domains such as sonar interpretation with the HASP/SIAP system (Nii et al., 1982), the architecture evolved to incorporate more sophisticated control strategies. Inspired by this paradigm, we adapt the blackboard architecture for multi-agent communication: rather than a central controller assigning tasks, all agents operate autonomously, responding to requests posted on the blackboard. A central main agent then leverages the information contributed by sub-agents to solve the problem.

# 6  CONCLUSIONS & FUTURE WORK

We addressed the critical challenge of data discovery in large, heterogeneous data lakes, a key bottleneck for applying LLMs in data science. We introduced a novel multi-agent communication paradigm based on the blackboard architecture, which replaces rigid centralized task assignment with a flexible, decentralized model of agent collaboration. Extensive experiments on three data science benchmarks demonstrate that our framework consistently outperforms strong baselines, including RAG and the master–slave paradigm, achieving up to 57% relative improvement in end-to-end task success and a 9% relative gain in data discovery accuracy. These results highlight the importance of communication architecture in multi-agent systems and establish the blackboard paradigm as a scalable, flexible, and effective solution for complex data science workflows.

Future work could extend the blackboard architecture and paradigm beyond data science, as the proposed approach is general and applicable to a wide range of multi-agent systems and domains. Another promising direction is to investigate more adaptive strategies for data partitioning among agents, enabling the system to better handle dynamic and evolving data environments. Ultimately, our findings point toward a broader path for developing more capable, scalable, and autonomous multi-agent AI systems for real-world data analysis applications.

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

# A  DATASETS AND PREPROCESSING

To the best of our knowledge, KramaBench (Lai et al., 2025) is the only publicly available dataset for data science problems that explicitly require data discovery to answer the questions. We adopt this dataset as one of our evaluation benchmarks in this paper.

To further investigate this problem, we repurpose two widely used datasets for data science tasks, DSBench (Jing et al., 2025) and DA-Code (Huang et al., 2024), which were not originally designed to include a data discovery phase. In their original form, each question in these datasets is paired with the specific data files required to answer it. To adapt them to our setting, we remove this direct mapping: the model is provided only with the question, while all files from the dataset are aggregated into a single data lake. The model must therefore first identify the relevant files within the data lake and then use them to solve the question.

**Filtering:**  we observed that not all questions in these datasets are suitable for the data discovery setting. For instance, some questions provide no hints about the characteristics of the files needed to answer them, while others simply ask for computing a statistic on a column without specifying sufficient information to identify the relevant file. To address this issue, we manually filter out such questions and retain only those that include adequate cues for discovering the appropriate files. We exclude questions that request performing a specific operation on a particular column of a data file when the column's meaning or semantics are insufficiently described. In such cases, it would be infeasible to accurately identify the target column within the data lake, given that multiple files may contain columns with the same name. Furthermore, questions that focus solely on the operation itself—assuming access to only a single file—are also excluded, as they lack sufficient contextual information for meaningful retrieval or reasoning about the data file that needs to be discovered from the data lake. Finally, since our goal is to study information discovery in data science, we also exclude questions that can be answered without accessing any data files, as these are general data science questions not relevant to any data files. After this filtering process, the resulting dataset statistics are reported in Table 3. Finally, we provide the example IDs corresponding to the data instances retained from each dataset (DSBench and DA-Code) in Appendix F.

Table 3: Statistics of the datasets used in our evaluation setup.

| Dataset | #Tasks | Size of data lake | #Clusters created by Gemini 2.5 Pro |
|---|---|---|---|
| KramaBench | 104 | 1746[17] | 27[18] |
|   - Archeology | 12 | 5 | 3 |
|   - Astronomy | 12 | 1556 | 8 |
|   - Biomedical | 9 | 7 | 2 |
|   - Environment | 20 | 37 | 4 |
|   - Legal | 30 | 136 | 4 |
|   - Wildfire | 21 | 23 | 6 |
| DSBench | 253 | 48 | 12 |
| DA-Code | 91 | 145 | 26 |

**Evaluation:**  To evaluate the programs generated by the system, we execute each program and assess its final output against the reference answer for the given question. For each dataset, we adopt its original evaluation methodology. Specifically, for KramaBench, we use the official evaluation script provided in their repository.[19]  For DA-Code, we similarly rely on the official evaluation script released in their repository.[20]  For DSBench, we follow the original evaluation protocol that

---

[17]Note that, in line with the original benchmark design, we construct a separate data lake for each subtask. However, the reported number of files corresponds to the total number of files aggregated across all subtasks in the benchmark.

[18]Note that, in line with the original benchmark design, we construct a separate data lake for each subtask. However, the reported number of clusters corresponds to the total number of clusters aggregated across all subtasks in the benchmark.

[19]Available at: https://github.com/mitdbg/KramaBench

[20]Available at: https://github.com/yiyihum/da-code

uses LLM-based judging: the generated programs output is compared to the reference answer using Gemini 2.5 Pro as the judge LLM, with the prompt shown in Figure 4.

Evaluation prompt for
DS-Bench

Please judge whether the generated answer is right or wrong. We require that the correct answer to the prediction gives a clear answer, not just a calculation process or a disassembly of ideas. The
The question is:
{question}.
The true answer is:
{answer}.
The predicted answer is:
{prediction}.
If the predicted answer is right, please output "True". Otherwise output "False". Don't output any other text content. You only can output "True" or "False" (without quotes).

Figure 4: Evaluation prompt used for DSBench dataset using LLM as the judge.

# B  AGENTS' PROMPTS

This section presents the prompts used by the agents and baselines in this paper. Figure 12 shows the prompt for clustering the data lake into multiple partitions based on file names. Figure 6 presents the prompt used by the main agent in the blackboard system. Figure 7 shows the prompt for the file agents. Figure 8 displays the prompt used by the search agent. Figure 9 presents the prompt for the main agent in the master–slave system, and Figure 10 shows the prompt used by the RAG agent.

---

**File Clustering Prompt**

You are an expert in classifying different files and directories into related clusters. You'll be given a list of file names and directories. Assume that we want to assign each cluster of files to a specific person to analyze and use them. Therefore, we need to group similar files and directories together to assign them to the same person. Your task is to classify the files and directories into clusters based on their names.

# Your input:
   - file addresses: a list of file addresses and names put in different directories.

# Your output: You should generate a valid json object in ```json ``` block with the following structure:
   - "clusters": a list of valid json objects each containing:
      - "name": the name of the cluster
      - "files": a list of file addresses and names that belong to this cluster. If you want to have whole directory, just write the directory address, otherwise write the file address. If you want to point to a specific file in a directory, write the file address.
      - "description": a short description of the cluster
      - "reason": a short reason why these files are grouped together

# Your task: You should classify the files and directories into clusters based on their names that are similar to each other. For example, some files might only be about different dates about the same topic, or some files might be about different aspects of the same topic, or some files might be about different versions of the same file. You should group them together based on their names and the context of the files. Note that you don't necessarily need to group all files together, some files might be left out if they don't fit into any cluster, they can form a cluster by themselves. The files are now grouped into directories sometimes, you can use these directories to help you group them together; however, the files that are currently in the same directory might not be related to each other, so you should not assume that all files in the same directory are related to each other. You should only group files together if they are related to each other based on their names and context. Note that for directories that are about the same topic or project but they are still very large (e.g., 50+ files), try to group them into smaller clusters based on their names and context.

# file addresses:
{addresses}

---

Figure 5: Prompt used by for clustering the files in data lakes into partitions.

**Main Agent Blackboard Prompt**

You are a capable data science agent whose task is to solve a given data science problem. This is a multi-step process and you don't need to complete all steps in one go. In the start, you will be given a data science problem that you need to solve. You need to solve this problem in multiple steps. In each step, you can select one action from a set of possible actions and execute it. Eventually, when you have the final solution to the problem, you can state this and end the process.

# Your input:
  - Problem: a data science problem that you need to solve. This problem is given to you in the beginning of the process and you need to solve it in multiple steps.

# Actions:
In each step, you can select one action from the following list of actions:
## Action Name: "request_help"
### Definition: In this action, you can broadcast a request, for example, to get the required data to solve the problem, or general information from web. Currently, the following requests are supported: {possible_requests}
### your output: You should generate a valid json object in ```json``` block, without anything before or after it, with the following structure:
  - "action": "request_help"
  - "request": a string that describes the request. This should be a description of your request and how they can help you in solving the problem. This request should be specific and not general. For example, if you need a specific data, you should describe the data you need, not asking what data is available. Be specific about what you need and why you need it.
  - "reason": a short reason why you think this request is needed.
### Response to this action: This will be a list of json response from other agents who can help you with your request. You can use this response to help you in solving the problem. You should read these responses carefully and trust them. You can use the responses to help you in solving the problem. For example, if you requested for data, you should follow the instructions in the response to load the data. If you requested for help from other agents, you should read their responses and use them to help you in solving the problem.
## Action Name: "plan"
### Definition: In this action, you can generate a plan to solve the problem. This plan should include the steps that you need to take to solve the problem.
### your output: You should generate a valid json object in ```json``` block, without anything before or after it, with the following structure:
  - "action": "plan"
  - "plan": a string that describes the plan to solve the problem. This should be a description of the steps that you need to take to solve the problem.
  - "reason": a short reason why you think this plan is needed.
### Response to this action: The user will acknowledge your plan and asks you to execute it.
## Action Name: "run_code"
### Definition: In this action, you can ask the system to run a code for you and provide you the output of the code. This action can be specificly useful when you need to try something out and see the output of the code. This can be helpful in case you need to install a library, or you need to run a code to see the output of it, or you need to run a code to check if it works as expected.
### your output: You should generate a valid json object in ```json``` block, without anything before or after it, with the following structure:
  - "action": "run_code"
  - "code": a valid python code that can be used to solve the problem.
  - "reason": a short reason why you think this code is needed.
### Response to this action: The system will run the code and provide you the output of the code.
## action Name: "reason"
### Definition: In this action, you can provide a reasoning and thinking step by step about a specific part of the problem. This can be useful when you need to think about a particular aspect of the problem and how to solve it.
### your output: You should generate a valid json object in ```json``` block, without anything before or after it, with the following structure:
  - "action": "reason"
  - "reasoning": a string that describes your reasoning and thinking step by step about a specific part of the problem. This should be a description of your reasoning and how you think about the problem.
  - "reason": a short reason why you think this reasoning is needed.
### Response to this action: The user will acknowledge your reasoning and asks you to continue with the next step.
## action Name: "answer"
### Definition: In this action, you can provide the final answer to the problem. This answer includes the final code you want to provide as the response to the problem and the breaking down of the problem into subtasks and how you solved each subtask. This action stops the process, thus, you should only use this action when you have the final answer to the problem.
### your output: You should generate a valid json object in ```json``` block, without anything before or after it, with the following structure:
  - "action": "answer"
  - "code": a valid python code that can be used to solve the problem. This code should be the final code that you want to provide as the response to the problem. It should load the data, preprocess it, and provide the final answer to the problem. In this code, you should include the response to each subtaks you have solved. You can use the print() function to print the answer to each subtask. For example, if you have an answer to subtask-1, subtask-2, and main-task (i.e., the final answer), you should print it like this:
print(json.dumps(
{{"subtask-1": answer1,
"subtask-2": answer2,
"main-task": answer
}}, indent=4))
You can find a suitable indentation for the print statement. Always import json at the beginning of your code. The output of this code will be used to evaluate the final answer to the problem, thus, make sure that the output is in a valid json format. Specifically, for the main task, just print the final answer to the problem.
  - "structured_response": a valid json object that contains the structured response to the problem. This should include the breaking down of the problem into subtasks and how you solved each subtask. This should be a valid json object that contains the following fields:
    - "id": str, that is always "main-task" for the main task. For each subtask, use "subtask-1", "subtask-2", etc.
    - "query": str, the question the step is trying to answer. Copy down the question from bellow for the main task.
    - "data_sources": list[str], the data sources you need to answer the question. Include all the file names you need for the main task.
    - "subtasks": list[dict], a list of subtasks. Each subtask should have the same structure as the main task.
  an example of this can be seen here: {example_json}
### Response to this action: The user will run the code and provide you the output of the code if there is any error. You should fix all errors even if they are warnings. If there is no error, the user will acknowledge your answer and end the process.

# Your task: This is a multi-step process and each step you should select one action and generate the output for that action. In response, the user will provide you the response to your action. You can use this response to help you in solving the problem. You can repeat this process until you have the final answer to the problem. When you have the final answer, you can use the "answer" action to provide the final answer to the problem.

Now, lets start the process for the following problem:
{query}

Figure 6: Prompt used by the main agent for the blackboard system.

File Agent
Prompt

You are a capable data scientist who is specialized in loading, analyzing, and cleaning data. You are responsible for handling a set of given files and directories. This is the list of files and directories you have to work with:
{files}

These files contain information about the following topics:
{topics}

Your name is: {name}

This is a multi-step process and you don't need to complete all steps in one go. Here I will explain the whole process:

## step 1: Getting information about the files: In this step, you have a chance to request for accessing a part of some of the files. To do this, you should generate a valid json list in ```json``` block that contains the address to the files you want me to give you a data sample from. for example Your output should be like this, without any additional text or explanation:
```json
["file1.txt", "file5.txt"]
```
Note that in cases where you can guess how other files look like based on a few of them, you don't need to request for all of them. Specifically, when the only difference between files is the file name is based on date, you can just request for one or a few of them and assume that the rest of them are similar. However, the file names are different and have different formats, you should request for all of them. Based on your request, I will give you a sample of the files. For example, for csv files, I will give you a few rows of the data (that might not be loaded correctly, so you should be careful about that), and for json files, I will give you a few objects from the file. For other formats, I will give you a few starting lines of the file.

## step 2: Analysing the data and how to load and clean it: When the data is loaded and given to you, you should analyze the fields in the data, how it should be effectively loaded, and how it should be cleaned. Specifically, the data should be cleaned for a data science problem, thus, some preprocessing steps should be done. For example, if the data contains missing values, you should decide how to handle them, or if the data contains na values, you should decide how to handle them. Additionally, be able to figure out what each column or row in the data means and you should be able to provide a description of them. Moreover, how combining data from multiple files can help in answering the question should be considered. This step happens when I provide you the data samples from the previous step.

## step 3: Checking if the data can be used to answer a question or a part of it: This step is like a loop and may occur multiple times. In this step, I provide you a request about a data access problem for answering a question. You should check if the data you have from each file or by combining data from multiple files can help in answering the question or a part of it. Your output for this step should be a valid json object in ```json``` block that contains the following fields:
  - "agent_name": your name, which is the same as the name you provided in the beginning of the process.
  - "can_help": a boolean value that indicates if the data you have can help in answering the question or data access request or a part of it. You can combine data from multiple files to answer the question or a part of it. If you think the data can help, set this to true, otherwise set it to false.
  - "reason": a short reason why you think the data can help or not.
  - "code": a valid python code that can be used to load the data and preprocess it in a way to be useful for fullfiling the request. This code should be able to load the data and preprocess and clean (e.g., dropping rows or columns that are nor part of the data) it in a way that it can be used to answer the question or a part of it. You can use any python library you want, but you should be able to explain why you are using it. If you use a library that is not installed by default, you should comment it in the code and explain why you need it and how to install it. In this code, use the full file addresses to load the data, not just the file names. For example, if the file is in a directory called "data", you should use "data/file.csv" instead of just "file.csv". If "can_help" is false, this can should be an empty string.
  - "data_explanation": a short explanation of the data, e.g, what each column or row means, what the data is about, etc. This should be a short explanation of the data that can help in understanding the data and how to use it.
  - "data_sample": a small sample of the data that can help in understanding the data and how to use it. Here you can provide a few rows, the column types and names, or a few objects from the data. This should be a small sample of the data that can help in understanding the data and how to use it. For example, if the data is a csv file, you can provide a few rows of the data, or if the data is a json file, you can provide a few objects from the data, or if it is a text file, provide a few rows. if "can_help" is false, this can should be an empty string.
  - "libraries": a list of libraries that we need to install in order to run the code. This should be a list of libraries that are not installed by default and you need to install them in order to run the code. If you don't need any additional libraries, you can leave this field empty.
  - "necessary_steps": a list of the steps that is necessary to take in order to correctly load and preprocess the data, For example, if the cvs file has a header, you should mention that in this list. Another example is if the header starts from a specific row, you should mention that in this list. Another example if the data has missing values, you should mention that in this list.

Now, lets start the process with the first step.

Figure 7: Prompt used by the file agent for the both master-slave and blackboard system.

**Search Agent Prompt**

You are a capable data science agent whose task is to search for information on the web. Your name is "{name}". Your goal is to find relevant information about the request that the user has provided. This is a multi-step process and you don't need to complete all steps in one go. In the start, you will be given a request about some sort of information access problem that you need to solve. In order to solve response to the request, you need to follow these steps:

# steps
## step 1: Analyzing the request and checking if you can help: In this step, the user provides you with a request that explains its information need. You should analyze the request and check if you can help in solving the request. You are only able to help in requests that are about finding information from web. You cannot help with requests that are about finding information from files or databases.
### input to this step:
  - request: a string that describes the request from the user. This request explains the information need of the user and what they are looking for.
### output of this step: You should generate a valid json object in ```json``` block, without anything before or after it, with the following structure:
  - "can_help": a boolean value that indicates if you can help in solving the request or not. If you can help, set this to true, otherwise set it to false. You cannot help in requests that are about accessing files or datasets, or requests that are not directly about searching information on the web. You can only help in finding information that is not related to datasets. You can help with libraries, tools, or general information that is not related to datasets.
  - "reason": a short reason why you think you can help or not. If you can help, explain why you think you can help.
### response to this step: The user will acknowledge your response and ask you to continue with the next step if you can help.

## step 2: Generating a search query: In this step, you need to generate a list of search queries that can be used to search for information on the web. You should generate a list of queries that are relevant to the request and can help in finding the information the user is looking for.
### input to this step:
  - request: a string that describes the request from the user. This request explains the information need of the user and what they are looking for.
### output of this step: You should generate a valid json object in ```json``` block, without anything before or after it, with the following structure:
  - "queries": a list of strings that describes the search queries that can be used to search for information on the web. These queries should be relevant to the request and can help in finding the information the user is looking for. You can generate multiple queries if you think they are relevant to the request.
  - "reason": a short reason why you think these queries are relevant to the request and can help in finding the information the user is looking for.
### response to this step: In response, the user will provide the results of the search queries you generated. You should read these results carefully. Then we go to the next step.

## step 3: Analyzing the search results: In this step, you need to analyze the search results and check if they are relevant to the request. You should check if the search results contain the information the user is looking for. If they do, you should extract the relevant information from the search results and provide it to the user. Otherwise, you can generate a new search query and go back to step 3.
### input to this step:
  - search_results: a list of search results that were returned from the search queries you generated in the previous step.
### output of this step: You should generate a valid json object in ```json``` block, without anything before or after it, with the following structure:
  - "stop_search": a boolean value that indicates if you should stop searching or not. If you found the information the user is looking for, set this to true, otherwise set it to false. Remember that you have a limited search budget. Thus, when you are informed that your search budget is over, you should stop searching and provide the information you found so far. You can stop searching even before your budget is over if you think you found the information the user is looking for.
  - "queries": a list of strings that describes the search queries that can be used to search for information on the web. These queries should be relevant to the request and can help in finding the information the user is looking for. You can generate multiple queries if you think they are relevant to the request. If you found the information the user is looking for, this should be an empty list.
  - "response_to_request": a string that describes the response to the request. This should be a description of the information you found in the search results that is relevant to the request. If you found the information the user is looking for, this should contain the relevant information. If you didn't find any relevant information, this should be an empty string. If you don't want ot stop searching, this should be an empty string.
  - "reason": a short reason why you think you should stop searching or not. If you found the information the user is looking for, explain why you think you found it. If you didn't find any relevant information, explain why you think you didn't find it. If you don't want to stop searching, explain why you think you should continue searching.
### response to this step: If you stopped searching, the user will acknowledge your response and end the process. If you didn't stop searching, the user will provide you with the search results for the new queries you generated in the previous step. Then we go back to step 3 and continue the process.

Now, lets start the process with the first step.

request: {request}

Figure 8: Prompt used by the search agent for the, master-slave, RAG, and blackboard system.

**Main Agent Master-Slave Prompt**

You are a capable data science agent whose task is to solve a given data science problem. This is a multi-step process and you don't need to complete all steps in one go. In the start, you will be given a data science problem that you need to solve. You need to solve this problem in multiple steps. In each step, you can select one action from a set of possible actions and excute it. Eventually, when you have the final solution to the problem, you can state this and end the process.

# Your input:
   - Problem: a data science problem that you need to solve. This problem is given to you in the beginning of the process and you need to solve it in multiple steps.

# Actions:
In each step, you can select one action from the following list of actions:
## Action Name: "request_data"
### Definition: In this action, you can select one of the agents who is responsible for loading and preprocessing the data to help you with providing the data you need to solve the problem. You can request for a specific data or a specific part of the data. Currently, you can call the following agents to help you with your request: {possible_requests}
### your output: You should generate a valid json object in ```json``` block, without anything before or after it, with the following structure:
   - "action": "request_data"
   - "agent_name": the name of the agent you want to request data from. This should be one of the agents who is responsible for loading and preprocessing the data.
   - "request": a string that describes the request. This should be a description of your request and how they can help you in solving the problem. This request should be specific and not general. For example, if you need a specific data, you should describe the data you need, not asking what data is available. Be specific about what you need and why you need it.
   - "reason": a short reason why you think this request is needed.
### Response to this action: This will be a json object from the agent. You can use this response to help you in solving the problem. You should read the response carefully and trust it. You can use the responses to help you in solving the problem. For example, if you requested for data, you should follow the instructions in the response to load the data. If you requested for help from other agents, you should read their responses and use them to help you in solving the problem.
## Action Name: "plan"
### Definition: In this action, you can generate a plan to solve the problem. This plan should include the steps that you need to take to solve the problem.
### your output: You should generate a valid json object in ```json``` block, without anything before or after it, with the following structure:
   - "action": "plan"
   - "plan": a string that describes the plan to solve the problem. This should be a description of the steps that you need to take to solve the problem.
   - "reason": a short reason why you think this plan is needed.
### Response to this action: The user will acknowledge your plan and asks you to execute it.
## Action Name: "run_code"
### Definition: In this action, you can ask the system to run a code for you and provide you the output of the code. This action can be specifically useful when you need to try something out and see the output of the code. This can be helpful in case you need to install a library, or you need to run a code to see the output of it, or you need to run a code to check if it works as expected.
### your output: You should generate a valid json object in ```json``` block, without anything before or after it, with the following structure:
   - "action": "run_code"
   - "code": a valid python code that can be used to solve the problem.
   - "reason": a short reason why you think this code is needed.
### Response to this action: The system will run the code and provide you the output of the code.
## action Name: "reason"
### Definition: In this action, you can provide a reasoning and thinking step by step about a specific part of the problem. This can be useful when you need to think about a particular aspect of the problem and how to solve it.
### your output: You should generate a valid json object in ```json``` block, without anything before or after it, with the following structure:
   - "action": "reason"
   - "reasoning": a string that describes your reasoning and thinking step by step about a specific part of the problem. This should be a description of your reasoning and how you think about the problem.
   - "reason": a short reason why you think this reasoning is needed.
### Response to this action: The user will acknowledge your reasoning and asks you to continue with the next step.
## action Name: "answer"
### Definition: In this action, you can provide the final answer to the problem. This answer includes the final code you want to provide as the response to the problem and the breaking down of the problem into subtasks and how you solved each subtask. This action stops the process, thus, you should only use this action when you have the final answer to the problem.
### your output: You should generate a valid json object in ```json``` block, without anything before or after it, with the following structure:
   - "action": "answer"
   - "code": a valid python code that can be used to solve the problem. This code should be the final code that you want to provide as the response to the problem. It should load the data, preprocess it, and provide the final answer to the problem. In this code, you should include the response to each subtask you have solved. You can use the print() function to print the answer to each subtask. For example, if you have an answer to subtask-1, subtask-2, and main-task (i.e., the final answer), you should print it like this:
print(json.dumps(
{{"subtask-1": answer1,
"subtask-2": answer2,
"main-task": answer
}}, indent=4))
You can find a suitable indentation for the print statement. Always import json at the beginning of your code. The output of this code will be used to evaluate the final answer to the problem, thus, make sure that the output is in a valid json format. Specifically, for the main task, just print the final answer to the problem.
   - "structured_response": a valid json object that contains the structured response to the problem. This should include the breaking down of the problem into subtasks and how you solved each subtask. This should be a valid json object that contains the following fields:
      - "id": str, that is always "main-task" for the main task. For each subtask, use "subtask-1", "subtask-2", etc.
      - "query": str, the question the step is trying to answer. Copy down the question from bellow for the main task.
      - "data_sources": list[str], the data sources you need to answer the question. Include all the file names you need for the main task.
      - "subtasks": list[dict], a list of subtasks. Each subtask should have the same structure as the main task.
   an example of this can be seen here: {example_json}
### Response to this action: The user will run the code and provide you the output of the code if there is any error. You should fix all errors even if they are warnings. If there is no error, the user will acknowledge your answer and end the process.

# Your task: This is a multi=step process and each step you should select one action and generate the output for that action. In response, the user will provide you the response to your action. You can use this response to help you in solving the problem. You can repeat this process until you have the final answer to the problem. When you have the final answer, you can use the "answer" action to provide the final answer to the problem.

Now, lets start the process for the following problem:
{query}

Figure 9: Prompt used by the main agent for the master-slave system.

**Main Agent RAG Prompt**

You are a capable data science agent whose task is to solve a given data science problem. This is a multi-step process and you don't need to complete all steps in one go. In the start, you will be given a data science problem that you need to solve. You need to solve this problem in multiple steps. In each step, you can select one action from a set of possible actions and excute it. Eventually, when you have the final solution to the problem, you can state this and end the process.

# Your input:
  - Problem: a data science problem that you need to solve. This problem is given to you in the beginning of the process and you need to solve it in multiple steps.
  - top_k_relevant_docs: a list of address and snippets from top-k relevant files that can be used to answer the problem. You should use these files to answer the problem. These files are provided to you in the beginning of the process and you can use them to answer the problem.

# Actions:
In each step, you can select one action from the following list of actions:
## Action Name: "search"
### Definition: In this action, you can generate a search query to search for information on the web. This can be useful when you need to find more information about the problem or the data you have.
### your output: You should generate a valid json object in ```json``` block, without anything before or after it, with the following structure:
  - "action": "search"
  - "request": a string that describes the search query. This should be a description of your request and how they can help you in solving the problem. This request should be specific, and not general. For example, if you need a specific data, you should describe the data you need, not asking what data is available. Be specific about what you need and why you need it.
  - "reason": a short reason why you think this request is needed.
### Response to this action: This will be a list of relevant information from the web that can help you in solving the problem. You should read these responses carefully and trust them. You can use the responses to help you in solving the problem.
## Action Name: "plan"
### Definition: In this action, you can generate a plan to solve the problem. This plan should include the steps that you need to take to solve the problem.
### your output: You should generate a valid json object in ```json``` block, without anything before or after it, with the following structure:
  - "action": "plan"
  - "plan": a string that describes the plan to solve the problem. This should be a description of the steps that you need to take to solve the problem.
  - "reason": a short reason why you think this plan is needed.
### Response to this action: The user will acknowledge your plan and asks you to execute it.
## Action Name: "run_code"
### Definition: In this action, you can ask the system to run a code for you and provide you the output of the code. This action can be specificly useful when you need to try something out and see the output of the code. This can be helpful in case you need to install a library, or you need to run a code to see the output of it, or you need to run a code to check if it works as expected.
### your output: You should generate a valid json object in ```json``` block, without anything before or after it, with the following structure:
  - "action": "run_code"
  - "code": a valid python code that can be used to solve the problem.
  - "reason": a short reason why you think this code is needed.
### Response to this action: The system will run the code and provide you the output of the code.
## action Name: "reason"
### Definition: In this action, you can provide a reasoning and thinking step by step about a specific part of the problem. This can be useful when you need to think about a particular aspect of the problem and how to solve it.
### your output: You should generate a valid json object in ```json``` block, without anything before or after it, with the following structure:
  - "action": "reason"
  - "reasoning": a string that describes your reasoning and thinking step by step about a specific part of the problem. This should be a description of your reasoning and how you think about the problem.
  - "reason": a short reason why you think this reasoning is needed.
### Response to this action: The user will acknowledge your reasoning and asks you to continue with the next step.
## action Name: "answer"
### Definition: In this action, you can provide the final answer to the problem. This answer includes the final code you want to provide as the response to the problem and the breaking down of the problem into subtasks and how you solved each subtask. This action stops the process, thus, you should only use this action when you have the final answer to the problem.
### your output: You should generate a valid json object in ```json``` block, without anything before or after it, with the following structure:
  - "action": "answer"
  - "code": a valid python code that can be used to solve the problem. This code should be the final code that you want to provide as the response to the problem. It should load the data, preprocess it, and provide the final answer to the problem. In this code, you should include the response to each subtaks you have solved. You can use the print() function to print the answer to each subtask. For example, if you have an answer to subtask-1, subtask-2, and main-task (i.e., the final answer), you should print it like this:
print(json.dumps(
{{"subtask-1": answer1,
"subtask-2": answer2,
"main-task": answer
}}, indent=4))
You can find a suitable indentation for the print statement. Always import json at the beginning of your code. The output of this code will be used to evaluate the final answer to the problem, thus, make sure that the output is in a valid json format. Specifically, for the main task, just print the final answer to the problem.
  - "structured_response": a valid json object that contains the structured response to the problem. This should include the breaking down of the problem into subtasks and how you solved each subtask. This should be a valid json object that contains the following fields:
    - "id": str, that is always "main-task" for the main task. For each subtask, use "subtask-1", "subtask-2", etc.
    - "query": str, the question the step is trying to answer. Copy down the question from bellow for the main task.
    - "data_sources": list[str], the data sources you need to answer the question. Include all the file names you need for the main task.
    - "subtasks": list[dict], a list of subtasks. Each subtask should have the same structure as the main task.
  an example of this can be seen here: {example_json}
### Response to this action: The user will run the code and provide you the output of the code if there is any error. You should fix all errors even if they are warnings. If there is no error, the user will acknowledge your answer and end the process.

# Your task: This is a multi=step process and each step you should select one action and generate the output for that action. In response, the user will provide you the response to your action. You can use this response to help you in solving the problem. You can repeat this process until you have the final answer to the problem. When you have the final answer, you can use the "answer" action to provide the final answer to the problem.

Now, lets start the process for the following problem:
{query}

Relevant files:
{top_k_relevant_docs}

Figure 10: Prompt used by the main agent for the RAG system.

## C    IMPLEMENTATION DETAILS

**Presenting Files to File Agents:**    A file agent may request a file by name, in which case it is shown a subset of the files contents. For this case, we employ a controlled procedure for loading and presenting the data to the agent, as described below:

- Files with *.csv* format: In this case, we use the *pandas*[21] library to load the CSV files, presenting the column names, their data types, and the top 20 rows of the table to the agent.

- Files with *.gpkg* format: which provides a pandas-like interface for geospatial data. The agent is then presented with the column names, their data types, and the top 20 rows of the table.

- Files with *.xlsx* format: In this case, we use the *pandas*[22] library to handle this file format. For files containing multiple sheets, we provide the agent with all sheet names, the data types of columns in each sheet, and the top 20 rows from each sheet.

- Files with *.npz* format: In this case, we utilize the *numpy*[23] library to load the data. The agent is then presented with all keys and their corresponding values within this data structure.

- Files with *.cdf* format: In this case, we utilize the *cdflib*[24] library to load the data. For presentation, we call the `cdf_info` and `globalattsget` functions on the loaded data structure, concatenate their outputs, and provide the result to the agent.

- Any other data format: In this case, we open the files using Pythons `open` function and present the first 20 lines of the file to the agent.

**Inference Setup.**    We limit the maximum number of actions taken by the main agent to $T = 10$. For decoding, we use nucleus sampling (Holtzman et al., 2020) with a temperature of $\tau = 0.1$. Proprietary models are accessed through Vertex AI,[25] while open-source models are served using the vLLM library.[26] At each generation step, we cap the output length at 8,192 tokens. We evaluate three proprietary LLMs—Gemini 2.5 Pro, Gemini 2.5 Flash (Gemini-Team, 2025), and Claude 4 Opus (Anthropic, 2025)—alongside an open-source model specialized for code generation, Qwen3-Coder-30B-A3B-Instruct (Qwen-Team, 2025).[27] Experiments with open-source models are conducted on 2 NVIDIA A100 GPUs (80GB VRAM each) with 128GB RAM.

---

[21]Available at: `https://pandas.pydata.org/`
[22]Available at: `https://pandas.pydata.org/`
[23]Available at: `https://numpy.org/`
[24]Available at: `https://cdflib.readthedocs.io/en/latest/`
[25]`https://cloud.google.com/vertex-ai?hl=en`
[26]`https://docs.vllm.ai/en/latest/`
[27]`https://huggingface.co/Qwen/Qwen3-Coder-30B-A3B-Instruct`

## D EXAMPLES AND CASE STUDIES

This section presents several case studies highlighting different aspects of the Blackboard system.

Figure 11 illustrates an example where a file agent requests access to files and performs their analysis. Figures 13 and 14 illustrate scenarios where the main agent lacked domain-specific knowledge and therefore posted requests on the blackboard seeking relevant information. In these cases, the search agent contributed by retrieving the necessary knowledge from the web, enabling the system to proceed with problem solving, which shows the effectiveness of search agent in problem solving.

Another example of a blackboard request is shown in Figure 15. In this example, specifically file agents responded to the request. Here, the main agent, given a data science question, formulated a request specifying the likely column names and data formats required, along with guidance for interpretation. In response, three out of eight helper agents contributed. Although the relevant files were spread across different clusters managed by separate file agents, each responding agent independently provided file addresses, code snippets for loading the data, explanations of the structure, and suggested preprocessing steps. Together, these contributions encompassed all the ground-truth files needed to solve the problem. This case demonstrates how the main agent can effectively leverage the blackboard to coordinate decentralized knowledge and achieve accurate data discovery.

In cases where none of the sub-agents can address a given request, as illustrated in Figure 17, the main agent attempts multiple reformulations of the query to facilitate the retrieval of relevant sub-information or related content. If these attempts remain unsuccessful, the agent resorts to alternative strategies, such as searching local files through code generation. If it still fails to locate or infer the necessary information from the partially gathered evidence, it ultimately terminates the search.

Figure 16 compares programs generated by the Blackboard and Master–Slave systems. The Blackboard agent produced the correct solution by accurately interpreting the prompt and selecting the appropriate data files. Specifically, it identified that the patients `Age` was located in `mmc1.xlsx` and that the requested `APP-Z score` was in `mmc7.xlsx`. By contrast, the Master–Slave agent misinterpreted the request and instead used a general protein abundance score (`APP_log2_abundance`) from the wrong file, `mmc2.xlsx`. This misstep resulted in an incorrect answer of 74, whereas the Blackboard agents precise data discovery and reasoning yielded the correct answer of 60.

**User**

Initial Prompt (not shown to save space)

**File Agent**

```` ```json\n[\n \"worldcities.csv\",\n \"roman_cities.csv\"\n]\n``` ````

**User**

This is a sample of the files you requested to see:
File name: worldcities.csv

Column data types (might be loaded incorrectly and you need to fix them):
city        object\ncity_ascii    object\nlat      float64\nlng      float64\ncountry      object\niso2      object\niso3      object\nadmin_name
object\ncapital      object\npopulation   float64\nid        int64
Table: [removed to save space]
File name: roman_cities.csv
Column data types (might be loaded incorrectly and you need to fix them):
Primary Key        object\nAncient Toponym      object\nModern Toponym      object\nProvince      object\nCountry
object\nBarrington Atlas Rank      object\nBarrington Atlas Reference    object\nStart Date      int64\nEnd Date
float64\nLongitude (X)      float64\nLatitude (Y)      float64\nSelect Bibliography      object
Table: [removed to save space]

Now please do the second step in your process based on the given instructions and the files you have read.

**File Agent**

Based on the data samples, here is my analysis of the files and a plan for loading and cleaning them.\n\n### Data Analysis and Cleaning Strategy\n\n**1. File: `/usr/local/google/home/asalemi/Documents/karmabench-project/Kramabench/data/archeology/input/worldcities.csv`**\n\n* **Description:** This file contains a comprehensive list of modern cities across the world. Each entry includes the city's name, geographic coordinates (latitude and longitude), country information, administrative details, and population.\n* **Column Analysis:**\n * `city`, `city_ascii`: The name of the city. `city_ascii` is a clean, ASCII-compatible version, which is ideal for matching and processing.\n * `lat`, `lng`: Geographic coordinates, crucial for any spatial analysis.\n * `country`, `iso2`, `iso3`: Country information, useful for grouping and filtering.\n * `admin_name`: The name of the administrative division (e.g., state, province).\n * `capital`: Indicates if the city is a primary, admin, or minor capital. Contains missing values which likely mean the city is not a capital.\n * `population`: The estimated population of the city.\n * `id`: A unique identifier for each city.\n* **Loading and Cleaning Plan:**\n 1. Load the data using a library like Pandas.\n 2. The first column in the sample is an unnamed index from the CSV file; it should be ignored during loading by setting `index_col=0`.\n 3. The `capital` column has missing values. These can be filled with a placeholder like 'Not a capital' to make the data more explicit.\n 4. The `population` column is a float, which is appropriate. I will check for any `NaN` values and decide on a handling strategy if they exist.\n 5. No other major cleaning seems necessary from the sample, but a general check for missing values across all columns is a good practice.\n\n**2. File: `/usr/local/google/home/asalemi/Documents/karmabench-project/Kramabench/data/archeology/input/roman_cities.csv`**\n\n* **Description:** This file lists cities from the Roman era, providing their ancient and modern names, location, a rough timeline of existence, and references.\n* **Column Analysis:**\n * `Primary Key`: A unique identifier for each record.\n * `Ancient Toponym`: The historical name of the city.\n * `Modern Toponym`: The city's modern name. This is a key column for potentially linking with the `worldcities.csv` dataset. It contains 'Unknown' values.\n * `Province`, `Country`: Historical province and modern country.\n * `Start Date`, `End Date`: Approximate start and end years of the city's prominence. `End Date` has missing values, which might imply the city's continuity or that the date is unknown.\n * `Longitude (X)`, `Latitude (Y)`: Geographic coordinates. The column names are inconsistent with the other file and should be standardized.\n * `Select Bibliography`: Reference information, likely not needed for quantitative analysis but useful for sourcing.\n* **Loading and Cleaning Plan:**\n 1. Load the data using Pandas, ignoring the first unnamed index column (`index_col=0`).\n 2. Rename columns for consistency and ease of use, especially `Longitude (X)` to `lng` and `Latitude (Y)` to `lat`.\n 3. In the `Modern Toponym` column, replace the string 'Unknown' with a standard `NaN` (Not a Number) value to facilitate better data handling and filtering.\n 4. The `End Date` column contains missing values. These should be kept as `NaN`s, as imputing them could be misleading. A missing end date is informative in itself.\n\n### Combined Analysis\n\nBy combining these two datasets, we can create a powerful resource for historical and urban studies. The primary method for merging would be to join `roman_cities.csv` and `worldcities.csv` on their city names (`Modern Toponym` and `city_ascii`, respectively). This would allow us to:\n* Find the modern population of ancient Roman cities.\n* Analyze the geographical distribution of ancient cities that are still inhabited today.\n* Compare the administrative importance (e.g., `capital` status) of cities over time.\n\nA secondary, more robust merging strategy could be a **spatial join**. By comparing the coordinates from both files, we could identify ancient and modern cities that occupy the same location, even if their names have changed or are spelled differently. This would be particularly useful for overcoming the limitations of name-based matching.

Figure 11: An example analyzing files by the file agent.

An example of clustering data lake with Gemini 2.5 Pro

```json
{
    "clusters": [
        {
            "name": "NIFC Wildfire Statistics",
            "files": [
                "nifc_wildfires.csv",
                "nifc_lightning_caused_wildfires.csv",
                "nifc_human_caused_wildfires.csv",
                "nifc_lightning_caused_acres.csv",
                "nifc_human_caused_acres.csv",
                "nifc_suppression_costs.csv",
                "nifc_geographic_areas.gpkg"
            ],
            "description": "Comprehensive wildfire data from the National Interagency Fire Center (NIFC).",
            "reason": "All files are sourced from NIFC (as indicated by the 'nifc_' prefix) and provide detailed statistics on wildfires, including their causes (human vs. lightning), impact (acres burned), suppression costs, and geographic boundaries."
        },
        {
            "name": "NOAA Wildfire Data",
            "files": [
                "noaa_wildfires.csv",
                "noaa_wildfires_monthly_stats.csv",
                "noaa_wildfires_variabledescrip.csv"
            ],
            "description": "Wildfire data and metadata from the National Oceanic and Atmospheric Administration (NOAA).",
            "reason": "These files form a cohesive dataset from NOAA (as indicated by the 'noaa_' prefix), including the main data, monthly statistics, and a description of the variables used."
        },
        {
            "name": "State-Level Wildfire Statistics",
            "files": [
                "Wildfire_Acres_by_State.csv",
                "Wildfire_Fires_by_State.csv",
                "wildfire_total_fires_p45_54.csv"
            ],
            "description": "Aggregated statistics on the number of wildfires and acres burned, primarily categorized by state.",
            "reason": "These files all provide high-level, aggregated wildfire statistics, with a focus on state-by-state breakdowns as suggested by their names."
        },
        {
            "name": "Wildfire Environmental and Operational Data",
            "files": [
                "Fire_Weather_Data_2002-2014_2016.csv",
                "PublicView_RAWS_-3515561676727363726.csv",
                "annual_aqi_by_county_2024.csv",
                "cleaned_helicopter_requests_by_region.csv"
            ],
            "description": "Data related to the environmental conditions for wildfires and the operational response.",
            "reason": "This cluster groups files about factors influencing wildfires (weather data from RAWS), their consequences (air quality), and the operational response (helicopter usage)."
        },
        {
            "name": "US General Geographic and Demographic Data",
            "files": [
                "usa.gpkg",
                "NST-EST2024-POP.xlsx",
                "democratic_vs_republican_votes_by_usa_state_2020.csv",
                "state_abbreviation_to_state.json"
            ],
            "description": "General reference data for the United States, including geographic boundaries, population estimates, political data, and state name mappings.",
            "reason": "These files provide contextual geographic, demographic, and political information for the USA, which can be used to enrich the primary wildfire datasets."
        },
        {
            "name": "Utility Scripts",
            "files": [
                "load_data.py"
            ],
            "description": "A code script for data processing.",
            "reason": "This is a standalone Python script, likely used to load or manipulate the other data files in the project."
        }
    ]
}
```

Figure 12: An example of clustering data lake into partitions using Gemini 2.5 Pro.

**Question**

What is the age of the patient with the lowest APP-Z score?

**Main Agent**
**Request**

I need to find what 'APP-Z score' refers to in the context of the UCEC CPTAC3 dataset. The provided metadata file does not contain a column with this name. I need to know if this score is known by another name in the dataset or if it is derived from other columns. Please search for information on 'APP-Z score' and its relation to the CPTAC UCEC study.

**Search Agent**
**Response**

Based on the search results, 'APP-Z score' in the context of the CPTAC UCEC dataset does not refer to a specific column name but is a derived value. Here is a breakdown of the findings:\\n\\n1.  **'APP' stands for Acute Phase Protein:** The initial search results strongly indicated that 'APP' is an acronym for Acute Phase Protein. These are proteins whose plasma concentrations change in response to inflammation.\\n\\n2.  **'Z-score' refers to a normalization method:** The analysis of CPTAC data, particularly through portals like UALCAN, commonly uses z-score normalization. A z-score represents the number of standard deviations a data point is from the mean or median of a reference set (e.g., normal tissue or the entire cohort). This is a standard procedure in proteomics to compare protein expression levels across different samples.\\n\\n3.  **Specific APPs in the CPTAC UCEC Dataset:** The search identified specific Acute Phase Proteins that have been analyzed within the CPTAC UCEC dataset:\\n    *   **SAAL1 (Serum Amyloid A-like 1):** One study (aging-us.com) explicitly identifies SAAL1 as a major Acute Phase Protein and confirms that its protein levels were found to be significantly upregulated in UCEC tumors within the CPTAC database.\\n    *   **Bikunin (from the AMBP gene):** Another study (PMC9889512) identifies Bikunin as a positive Acute Phase Protein and analyzes its expression in the CPTAC UCEC dataset, noting a significant reduction in protein expression in tumors compared to normal tissue.\\n\\n**Conclusion:**\\nThe 'APP-Z score' is the z-score normalized expression value for a specific Acute Phase Protein. It is not a pre-existing column in the metadata but a calculated score. To find the source of this score, you should look for columns in your dataset corresponding to the gene or protein names of known Acute Phase Proteins, such as **SAAL1** or **AMBP** (the precursor gene for Bikunin). The z-score would then be calculated from the expression values in that column.

Figure 13: An example of the request by the main agent that the search agent has provided a guideline based on search results.

**Question**

what is the correlation between the proportion of generally unsafe air quality days according to the EPA and the amount of land affected by fires in 2024? Round to 2 decimal places.

**Main Agent**
**Request**

I need to find the definition of 'generally unsafe air quality days' according to the EPA. This information will help me understand what data to look for and how to process it to answer the main question.

**Search Agent**
**Response**

Based on the search results, the EPA does not use the exact phrase 'generally unsafe air quality days'. Instead, it uses the Air Quality Index (AQI) to classify air quality and its health implications. The term 'generally unsafe' would correspond to the AQI categories where the health effects apply to the general public, not just sensitive groups.\\n\\nAccording to the EPA's Air Quality Index (AQI) basics from AirNow.gov, the levels that are considered unhealthy for the general population are:\\n\\n* **Red (Unhealthy):** AQI value of 151 to 200. At this level, 'Some members of the general public may experience health effects; members of sensitive groups may experience more serious health effects.'\\n* **Purple (Very Unhealthy):** AQI value of 201 to 300. This is a 'Health alert: The risk of health effects is increased for everyone.'\\n* **Maroon (Hazardous):** AQI value of 301 and higher. This is a 'Health warning of emergency conditions: everyone is more likely to be affected.'\\n\\nTherefore, a 'generally unsafe air quality day' can be defined as a day when the AQI is 151 or higher.

Figure 14: An example of the request by the main agent that the search agent has provided a guideline based on search results.

**Question**

Which of the 10 NICC-defined geographic areas requested the most helicopters for firefighting?

**Main Agent Request**

I need access to data that can help me determine which of the 10 NICC-defined geographic areas requested the most helicopters for firefighting. Specifically, I am looking for datasets that contain information on firefighting resource requests, including the type of resource (e.g., helicopter), the requesting geographic area, and the date of the request.

| Utility Scripts | Search Agent Response | NIFC Wildfire Statistics | NIFC Wildfire Statistics | NOAA Wildfire Data |
|---|---|---|---|---|
| cannot help | cannot help | cannot help | cannot help | cannot help |

**State-Level Wildfire Statistics**

Yes, I can help. The provided files contain exactly the information requested. 'Wildfire_Acres_by_State.csv' includes state-level data on total acreage burned and population, and 'Wildfire_Fires_by_State.csv' contains the total number of fires per state. I can merge these to provide a comprehensive dataset.

**code**

```python
import pandas as pd
def load_and_process_data():
    acres_file = 'Wildfire_Acres_by_State.csv'
    fires_file = 'Wildfire_Fires_by_State.csv'
    df_acres = pd.read_csv(acres_file, index_col=0)
    df_fires = pd.read_csv(fires_file, index_col=0)
    df_acres['Total Acres Burned'] = df_acres['Total Acres Burned'].fillna(0)
    merged_df = pd.merge(df_acres, df_fires, on='State')
    return merged_df
```

**Data Explanation**

The processed data is a single table containing aggregated wildfire statistics for each US state. The columns include 'State', 'Total Acres Burned' (the total area burned by wildfires), 'Population' (the state's population), and 'Total Fires' (the total number of wildfire incidents).

**Data Sample**

State Total Acres Burned Population Total Fires\\n0    Alabama    20978.0    5157699    1525\\n1    Alaska    667075.0    740133    377\\n2    Arizona    282989.0    7582384    2191\\n3    Arkansas    25444.0    3088354    1219\\n4    California    1081144.0    39431263    8316\

**Suggested Steps**

1) Load 'Wildfire_Acres_by_State.csv' and 'Wildfire_Fires_by_State.csv' into pandas DataFrames. 2) When loading, specify `index_col=0` to correctly handle the unnamed index column in the source files. 3) Clean the acres data by filling the missing value for 'Total Acres Burned' for 'District of Columbia' with 0. 4) Merge the two DataFrames on the 'State' column to create a single, unified dataset.

**US General Geographic and Demographic Data**

I can fulfill the second part of your request. I have a dataset with the most recent population estimates for each US state. However, I do not have a dataset containing information about individual wildfires and their acreage.

**code**

```python
import pandas as pd
def load_population_data():
    file_path = '/NST-EST2024-POP.xlsx'
    df_pop = pd.read_excel(
        file_path,
        skiprows=3,
        skipfooter=1, # Skip the note at the bottom
        header=None
    )
    df_pop.columns = ['Geographic_Area', 'April_2020_Base', 'Pop_2020', 'Pop_2021', 'Pop_2022', 'Pop_2023', 'Pop_2024']
    df_pop = df_pop[df_pop['Geographic_Area'].str.startswith('.', na=False)]
    df_pop['Geographic_Area'] = df_pop['Geographic_Area'].str.lstrip('.')
    df_pop.rename(columns={'Geographic_Area': 'state'}, inplace=True)
    df_recent_pop = df_pop[['state', 'Pop_2024']].copy()
    df_recent_pop['Pop_2024'] = df_recent_pop['Pop_2024'].astype(int)
    return df_recent_pop
```

**Data Explanation**

This dataset contains the annual resident population estimates for US states from April 1, 2020, to July 1, 2024. The provided code extracts the state name and the most recent population estimate for 2024.

**Data Sample**

state    Pop_2024\\n0    Alabama    5157699\\n1    Alaska    740133\\n2    Arizona    7582384\\n3    Arkansas    3088354\\n4 California    39431263

**Suggested Libraries**

pandas, openpyxl

**Suggested Steps**

1) The Excel file must be loaded by skipping the first 3 rows which act as a header. 2) The footer row containing notes should be skipped. 3) Column names need to be assigned manually as they are spread across multiple rows in the original file. 4) Rows corresponding to national or regional summaries (e.g., 'United States', 'Northeast') must be filtered out. 5) The state name column contains a leading dot ('.') which must be removed for clean merging. 6) Select the state column and the column for the most recent year's population (2024).

**Wildfire Environmental and Operational Data**

Yes, I can help. The file `cleaned_helicopter_requests_by_region.csv` contains the exact information required: the total number of helicopter requests aggregated by NICC-defined geographic areas.

**code**

```python
import pandas as pd

def get_helicopter_requests_by_region():
    file_path = '/cleaned_helicopter_requests_by_region.csv'
    df = pd.read_csv(file_path, index_col=0)
    df_sorted = df.sort_values(by='Total Helicopter Requests', ascending=False)
    return df_sorted
```

**Data Explanation**

This dataset provides a summary of the total number of helicopter requests for firefighting, aggregated by geographic area. It contains two columns: 'Region', which lists the name of the NICC-defined geographic area, and 'Total Helicopter Requests', which is the count of requests for that region.

**Data Sample**

Region Total Helicopter Requests\\n2    Great Basin Area    232\\n6 Northwest Area    185\\n5    Northern Rockies Area    112\\n9 Southern California Area    89\\n4    Northern California Area    81\\n10    Southwest Area    65\\n8    Southern Area    49\\n7 Rocky Mountain Area    39\\n0    Alaska Area    19\\n1 Eastern Area    9\

**Suggested Steps**

1) Load the CSV file using the pandas library. 2) Set the `index_col=0` parameter during loading to correctly handle the unnamed index column in the file. 3) The data is already cleaned and aggregated, so no further preprocessing is needed. 4) Sort the DataFrame by 'Total Helicopter Requests' to easily identify the area with the most requests.

Figure 15: An example of the generated request by the blackboard system.

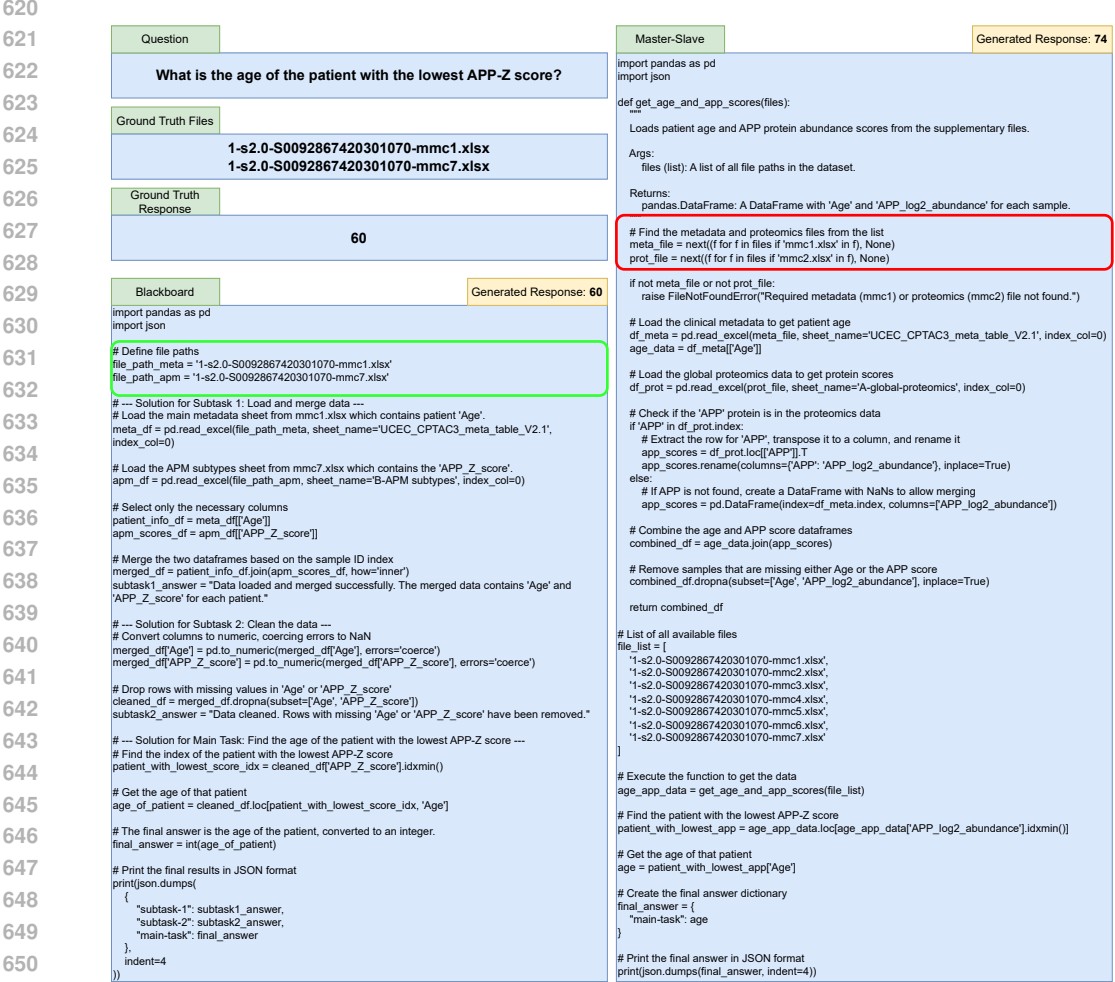

Figure 16: An example of the generated program by the blackboard system and master-slave system. The green box highlights where the blackboard system correctly selected the relevant files from the data lake, while the red box indicates where the master–slave system made an incorrect selection.

**Question**

What is the average Potassium in ppm from the first and last time the study recorded people in the Maltese area? Assume that Potassium is linearly interpolated between samples. Round your answer to 4 decimal places.

**Main Agent**
**Request**

I need access to study data that contains Potassium measurements in ppm from the Maltese area. The data should include timestamps or dates to identify when measurements were taken, and should indicate when people were recorded in the study. This data is needed to calculate the average Potassium levels from the first and last times people were recorded in the Maltese area.

None of the sub agents can help with this request

**Main Agent**
**Request**

I need to search for information about any available datasets or files that contain Potassium measurements in ppm from studies conducted in the Maltese area. Please search for any CSV files, Excel files, or data files that might contain this information along with temporal data showing when people were recorded in the study.

None of the sub agents can help with this request

**Main Agent**
**Run Code**

```python
import os
import glob

print("Files in current directory:")
for file in os.listdir('.'):
    print(f"  {file}")
print("\nLooking for data files:")
data_extensions = ['*.csv', '*.xlsx', '*.xls', '*.json', '*.txt', '*.data']
for ext in data_extensions:
    files = glob.glob(ext)
    if files:
        print(f"\n{ext} files found:")
        for file in files:
            print(f"  {file}")
```

Figure 17: An example of the requesting operation that none of the sub-agents can help with this request. In this case, the main agent retries multiple times with different reformulations of the request, and upon repeated failure, resorts to alternative actions, such as executing code to locate the relevant files autonomously.

**Question**

What is the estimation error, in terms of MAE, of NOAA SWPC's 3-day forecast of AP from March 9, 2025 for the period March 10-12, 2025?

| Action | Master-Agent |
|---|---|
| Requested Agent: Geomagnetic Forecasts | Request: I need to find the NOAA SWPC's 3-day forecast of the Ap index issued on March 9, 2025. Please provide me with the relevant file or a list of available forecast files so I can identify the correct one. |

Geomagnetic Forecasts denies to help because it doesn't have the knowledge

| Action | Master-Agent |
|---|---|
| Requested Agent: OMNI2 Interplanetary Data | Request: I need the observed daily Ap index for the dates March 10, 2025, to March 12, 2025. This is to compare with the forecasted values and calculate the estimation error. |

OMNI2 Interplanetary Data denies to help because it doesn't have the knowledge

| Action | Master-Agent |
|---|---|
| Requested Agent: STORM-AI Project Data | Request: I need the observed daily Ap index for the period of March 10, 2025, to March 12, 2025. This data is required to be compared against the forecasted Ap index values to calculate the forecast error. |

STORM-AI Project Data denies to help because it doesn't have the knowledge

| Action | Master-Agent |
|---|---|
| Search Agent | Search for a reliable source of observed daily planetary Ap index data. I need to find a downloadable data file (e.g., .txt or .csv format) or an API that provides historical and recent values. Please prioritize official sources like NOAA, GFZ Potsdam, or other space weather data centers. |

I have found several reliable, official sources for observed daily planetary Ap index data, including downloadable files and APIs, as requested.\\n\\nHere are the best sources:\\n\\n**1. GFZ German Research Centre for Geosciences (Potsdam)**\\nGFZ is the primary international body responsible for calculating and distributing the Kp and Ap indices. This is the most definitive source.\\n\\n*   **Direct Data Download (FTP):** You can find historical and recent data files in text format on their FTP server. The daily Ap values are typically found in files organized by year.\\n   *   **Link:** `ftp://ftp.gfz-potsdam.de/pub/home/obs/kp-ap/`\\n   *   **File Format Information:** A description of the data format is available here: `ftp://ftp.gfz-potsdam.de/pub/home/obs/kp-ap/wdc/wdc_fmt.txt`\\n*   **Web Portal and API:** Their main data portal provides interactive charts, data download forms, and access to a web service API.\\n   *   **Link:** `https://kp.gfz.de/en/`\\n\\n**2. NOAA's Space Weather Prediction Center (SWPC) and National Centers for Environmental Information (NCEI)**\\nNOAA provides easy access to this data, often referencing GFZ as the source.\\n\\n*   **Recent Data (Text and JSON):** The SWPC provides the last 30 days of K and A indices, as well as a JSON API for the most recent K-index values (from which Ap is derived).\\n   *   **Link:** `https://www.swpc.noaa.gov/products/planetary-k-index` (Check this page for links to the 30-day text file and the JSON data).\\n*   **Historical Data Files:** The NCEI hosts historical archives of geomagnetic indices, including Ap data from 1932 onwards.\\n   *   **Link:** `https://www.ncei.noaa.gov/products/geomagnetic-indices` (Look for the 'Kp, Ap, and Other Indices' section for download links).\\n\\nIn summary, for the most direct access to downloadable `.txt` files, the **GFZ FTP server** is the best option. For API access or more user-friendly web downloads, the **GFZ portal** and **NOAA SWPC** are excellent choices.

Figure 18: Example where the main controller in a Master–Slave system fails to route the query to the correct data sub-agent: after multiple retries, it falls back to the search agent, which returns only generic or incorrect information.

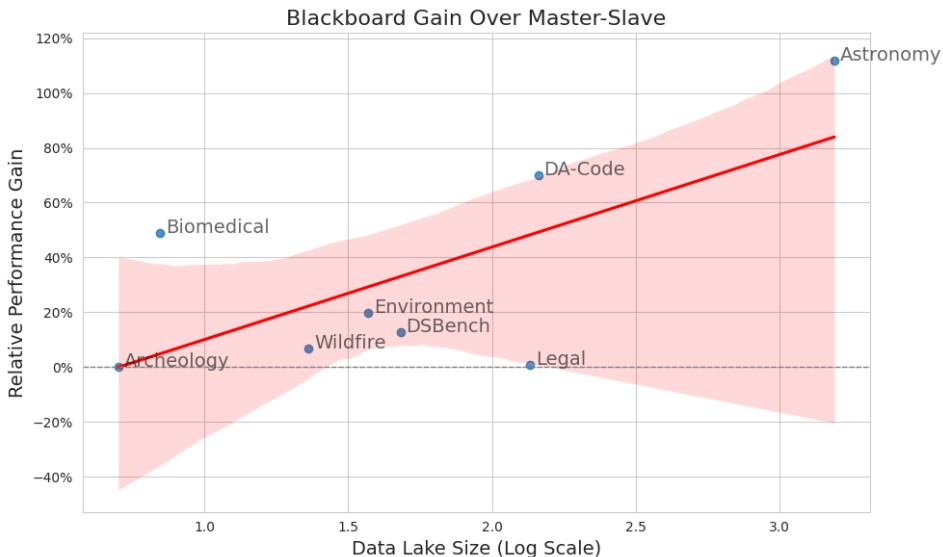

Figure 19: Relative performance gain of the Blackboard system over the Master–Slave system as a function of data lake size (log 10 scale). The red line represents the fitted regression line, with the shaded area showing the confidence interval. The results indicate a positive correlation between data lake size and the relative performance gain, suggesting that the Blackboard architecture scales more effectively with larger data lakes. This experiment is conducted using Gemini 2.5 Pro as the LLM.

# E FURTHER RESULTS ANALYSIS

**Scalability Comparison between Blackboard and Master–Slave Systems:** We report relative, rather than absolute, performance gains to ensure fair comparison across datasets with varying task difficulties and score ranges. Absolute improvements can be misleading when the baseline performance or achievable score range differs substantially—for instance, a 5-point gain might be minor in an easy task with high baseline scores but significant in a challenging one. Relative gain normalizes these differences by measuring proportional improvement with respect to the baseline, enabling consistent comparison across heterogeneous tasks. Using this normalized metric, we analyze the scalability of the Blackboard architecture compared to the Master–Slave baseline across datasets with different data lake sizes (Figure 19). Each point corresponds to a distinct task domain, and a fitted regression line reveals a clear positive correlation between data lake size and relative performance gain. This indicates that the Blackboard system scales more effectively as data environments grow larger, whereas the Master–Slave system exhibits limited scalability under such conditions.

**Runtime and Cost Analysis:** To assess the efficiency–cost trade-off of the Blackboard system relative to the RAG and Master–Slave baselines, we randomly sampled 50 questions from the KramaBench benchmark spanning all domains and measured both runtime and cost per question for each method. Unlike the RAG and Master–Slave baselines, which execute their component calls sequentially following the ReAct framework, the Blackboard architecture parallelizes sub-agent interactions: once the main agent posts a request to the shared blackboard, the corresponding sub-agents process it independently. As shown in Figure 20, the runtime of all three systems lies in a narrow band (132.0–145.2 seconds), indicating no significant difference in latency. In terms of monetary cost, the Blackboard system is more expensive per question (approximately $2.3\times$ the cost of RAG and $1.8\times$ that of Master–Slave), reflecting its increased token usage. However, this additional cost translates into substantial performance gains—54.1% over RAG and 18.8% over Master–Slave—so that Blackboard delivers markedly better answer quality while maintaining comparable runtime, offering a favorable accuracy–cost trade-off.

**Compare with Other More Advanced Data Science Baselines:** To further evaluate our method against advanced data-science-oriented agent frameworks, we compare the Blackboard system with

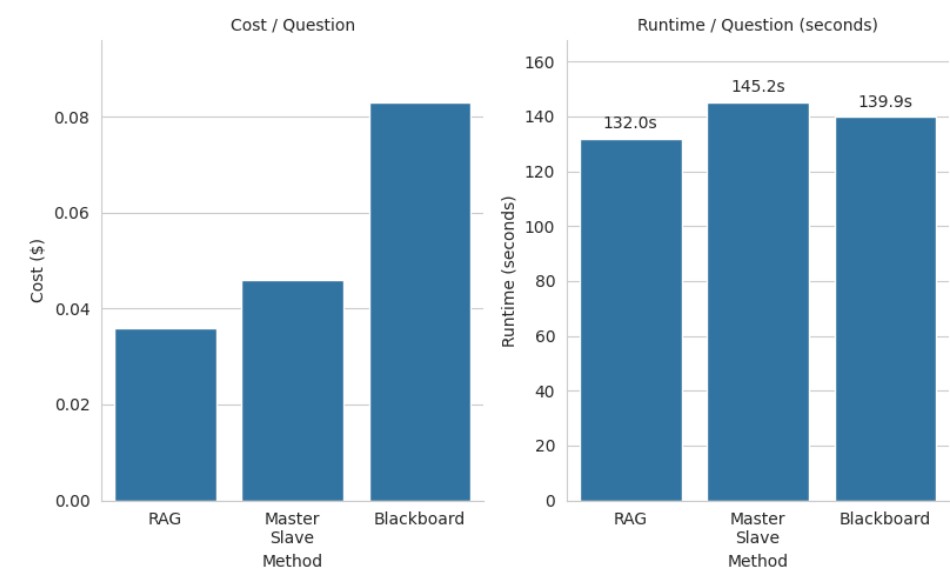

Figure 20: Runtime and cost analysis of the RAG, Master–Slave, and Blackboard systems on 50 examples from the KramaBench benchmark. We use Gemini 2.5 Pro as the LLM.

Table 4: Results on the KramaBench benchmark using Gemini 2.5 Pro as the underlying LLM.

| Method | Archaeology | Astronomy | Biomedical | Environment | Legal | Wildfire | Average |
|---|---|---|---|---|---|---|---|
| Data Interpreter | **41.67**% | 12.72% | 28.05% | 9.87% | 30.04% | 59.67% | 30.34% |
| AutoGen | 16.67% | 4.39% | 7.25% | 19.38% | 26.38% | 41.76% | 19.30% |
| Blackboard | 33.33% | **17.95**% | **36.83**% | **39.31**% | **34.92**% | **62.88**% | **37.53**% |

Data Interpreter (Hong et al., 2025) and AutoGen (Wu et al., 2023), both designed for multi-agent data analysis and reasoning. The results on the KramaBench benchmark are presented in Table 4. As shown, the Blackboard system outperforms these baselines in five out of six tasks as well as on the overall average, highlighting its stronger adaptability and coordination capabilities across diverse data-science scenarios.

**Effect of Clustering Based on File Name and Content and Number of Clusters on the Performance:** As described in Section 3, we employ Gemini 2.5 Pro to perform file clustering based on filenames. However, given that file contents can be lengthy, it is impractical to provide the full text of each file to the model for clustering. To address this, we additionally experiment with content-based clustering using a semantic embedding model. Specifically, we encode each file's content using E5-Large (Wang et al., 2022) and apply the KMeans algorithm (Lloyd, 1982) to partition the files into multiple clusters. Unlike Gemini, which automatically determines the number of clusters, KMeans requires this value to be specified. We evaluate settings with 2, 4, and 8 clusters and use datasets containing at least 100 files—namely, the Legal and Astronomy subsets from the KramaBench and DA-Code datasets.

The results of the content-based clustering are presented in Figure 21. Overall, increasing the number of clusters leads to higher performance, as it allows each sub-agent to specialize on a smaller subset of files and perform more targeted reasoning. Furthermore, to compare filename-based clustering with Gemini 2.5 Pro against content-based clustering using E5-Large embeddings, we report the results in Figure 22. As shown, clustering based on semantic content yields consistently better performance than clustering by filename alone. This indicates that our proposed framework generalizes effectively to embedding-based clustering and does not rely solely on large language models for grouping data files.

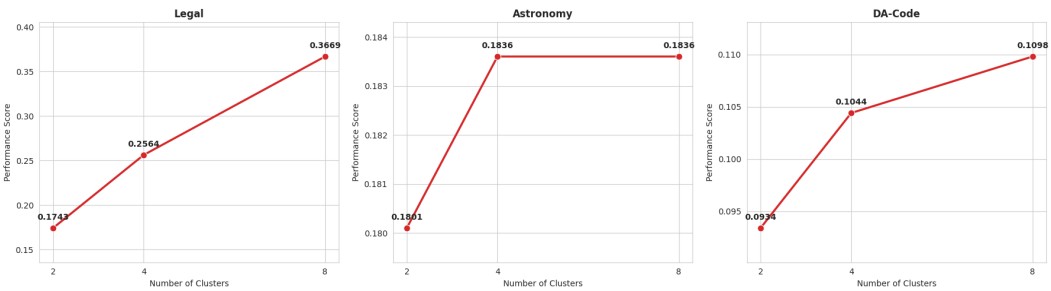

Figure 21: Effect of the number of clusters on the performance of the Blackboard system. Clustering is performed using the KMeans algorithm with E5-Large as the embedding model. We use Gemini 2.5 Pro as the LLM.

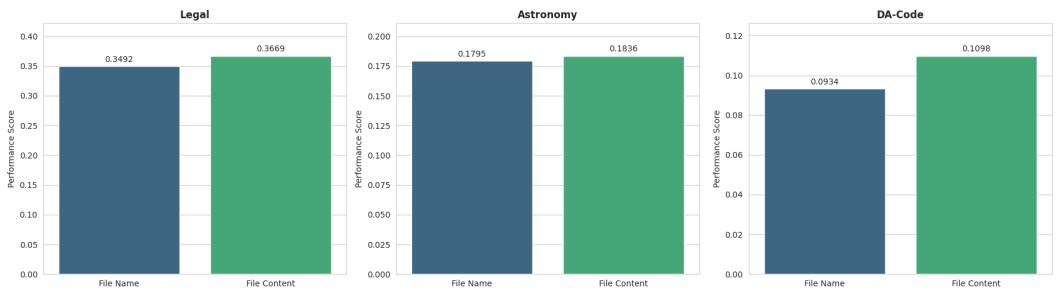

Figure 22: Effect of clustering based on filename using Gemini 2.5 Pro and based on file content using the KMeans algorithm with E5-Large as the embedding model. We use Gemini 2.5 Pro as the LLM.

## F    EXAMPLE IDS FOR FILTERED DATASETS

We retained the following example IDs from the instances in the DSBench dataset:

- 00000001_question17
- 00000001_question7
- 00000001_question16
- 00000001_question13
- 00000001_question12
- 00000001_question9
- 00000001_question10
- 00000001_question11
- 00000001_question18
- 00000001_question15
- 00000004_question50
- 00000004_question45
- 00000004_question42
- 00000004_question48
- 00000004_question47
- 00000004_question49
- 00000004_question44
- 00000004_question41
- 00000004_question43
- 00000005_question21
- 00000005_question29
- 00000005_question28
- 00000005_question24
- 00000005_question25

- 00000005_question23
- 00000005_question26
- 00000005_question20
- 00000005_question27
- 00000006_question17
- 00000006_question22
- 00000006_question21
- 00000006_question18
- 00000006_question24
- 00000006_question25
- 00000006_question19
- 00000006_question23
- 00000006_question26
- 00000006_question20
- 00000007_question17
- 00000007_question22
- 00000007_question7
- 00000007_question16
- 00000007_question6
- 00000007_question5
- 00000007_question13
- 00000007_question12
- 00000007_question21
- 00000007_question9

- 00000007_question10
- 00000007_question11
- 00000007_question3
- 00000007_question18
- 00000007_question4
- 00000007_question19
- 00000007_question23
- 00000007_question14
- 00000007_question2
- 00000007_question15
- 00000007_question20
- 00000007_question1
- 00000007_question8
- 00000008_question33
- 00000008_question31
- 00000008_question28
- 00000008_question32
- 00000008_question34
- 00000010_question17
- 00000010_question7
- 00000010_question16
- 00000010_question6
- 00000010_question5
- 00000010_question13

- 00000010_question12
- 00000010_question9
- 00000010_question10
- 00000010_question11
- 00000010_question3
- 00000010_question18
- 00000010_question4
- 00000010_question19
- 00000010_question14
- 00000010_question2
- 00000010_question15
- 00000010_question20
- 00000010_question1
- 00000010_question8
- 00000011_question7
- 00000011_question6
- 00000011_question5
- 00000011_question3
- 00000011_question4
- 00000011_question2
- 00000011_question1
- 00000011_question8
- 00000012_question7
- 00000012_question6
- 00000012_question5
- 00000012_question9
- 00000012_question10
- 00000012_question3
- 00000012_question4
- 00000012_question2
- 00000012_question1
- 00000012_question8
- 00000013_question17
- 00000013_question22
- 00000013_question16
- 00000013_question21
- 00000013_question18
- 00000013_question24
- 00000013_question19
- 00000013_question23
- 00000013_question20
- 00000016_question7
- 00000016_question6
- 00000016_question5
- 00000016_question12
- 00000016_question9
- 00000016_question11
- 00000016_question3
- 00000016_question2
- 00000016_question1
- 00000016_question8
- 00000017_question39
- 00000017_question33
- 00000017_question38
- 00000017_question29
- 00000017_question37
- 00000017_question30
- 00000017_question32
- 00000017_question35
- 00000017_question34
- 00000017_question36
- 00000018_question24
- 00000018_question25
- 00000018_question23
- 00000019_question6
- 00000019_question13
- 00000019_question9
- 00000019_question10
- 00000019_question14
- 00000019_question15
- 00000020_question33
- 00000020_question31
- 00000020_question29
- 00000020_question37
- 00000020_question30
- 00000020_question28
- 00000020_question32
- 00000020_question35
- 00000020_question34
- 00000020_question36
- 00000022_question39
- 00000022_question33
- 00000022_question38
- 00000022_question40
- 00000022_question37
- 00000022_question35
- 00000022_question34
- 00000022_question36
- 00000025_question39
- 00000025_question33
- 00000025_question38
- 00000025_question40
- 00000025_question37
- 00000025_question35
- 00000025_question41
- 00000025_question34
- 00000025_question36
- 00000027_question13
- 00000027_question12
- 00000027_question14
- 00000027_question15
- 00000029_question7
- 00000029_question6
- 00000029_question5
- 00000029_question9
- 00000029_question10
- 00000029_question3
- 00000029_question4
- 00000029_question2
- 00000029_question1
- 00000029_question8
- 00000030_question7
- 00000030_question6
- 00000030_question5
- 00000030_question3
- 00000030_question4
- 00000030_question2
- 00000030_question1
- 00000032_question7
- 00000032_question6
- 00000032_question5
- 00000032_question3
- 00000032_question4
- 00000032_question2
- 00000032_question1
- 00000033_question7
- 00000033_question6
- 00000033_question3
- 00000033_question4
- 00000033_question2
- 00000033_question1
- 00000033_question8
- 00000034_question7
- 00000034_question16
- 00000034_question6
- 00000034_question5
- 00000034_question13
- 00000034_question12
- 00000034_question9
- 00000034_question10
- 00000034_question11
- 00000034_question3
- 00000034_question4
- 00000034_question14
- 00000034_question2
- 00000034_question15
- 00000034_question1
- 00000034_question8
- 00000035_question7
- 00000035_question6
- 00000035_question5
- 00000035_question9
- 00000035_question3
- 00000035_question4
- 00000035_question2
- 00000035_question1
- 00000035_question8
- 00000038_question5
- 00000038_question3
- 00000038_question2
- 00000038_question1
- 00000043_question17
- 00000043_question7
- 00000043_question16
- 00000043_question6
- 00000043_question5
- 00000043_question13
- 00000043_question12
- 00000043_question9
- 00000043_question10
- 00000043_question11
- 00000043_question3
- 00000043_question18
- 00000043_question4
- 00000043_question19
- 00000043_question14
- 00000043_question2

- 00000043_question15
- 00000043_question20

- 00000043_question1
- 00000043_question8

We retained the following example IDs from the instances in the DA-Code dataset:

- di-text-001
- di-text-002
- di-text-003
- di-text-004
- di-text-005
- di-text-006
- di-text-007
- di-text-008
- di-text-009
- di-text-010
- di-text-011
- di-text-012
- di-text-013
- di-text-014
- di-text-015
- di-text-016
- di-text-017
- di-text-018
- di-text-019
- di-text-020
- di-text-021
- di-text-023
- di-text-024
- di-text-025
- di-text-027
- di-text-028
- di-text-029
- di-text-030
- di-text-031
- di-text-032
- di-text-033

- di-text-034
- di-text-035
- di-text-036
- di-text-037
- di-text-038
- di-text-039
- di-text-040
- di-text-041
- di-csv-001
- di-csv-002
- di-csv-003
- di-csv-005
- di-csv-006
- di-csv-007
- di-csv-008
- di-csv-009
- di-csv-010
- di-csv-011
- di-csv-012
- di-csv-013
- di-csv-014
- di-csv-015
- di-csv-016
- di-csv-017
- di-csv-018
- di-csv-019
- di-csv-020
- di-csv-021
- di-csv-022
- di-csv-023
- dm-csv-001

- dm-csv-002
- dm-csv-009
- dm-csv-010
- dm-csv-011
- dm-csv-016
- dm-csv-019
- dm-csv-020
- dm-csv-021
- dm-csv-028
- dm-csv-029
- dm-csv-032
- dm-csv-034
- dm-csv-038
- dm-csv-041
- dm-csv-049
- dm-csv-050
- dm-csv-055
- data-sa-011
- data-sa-012
- data-sa-022
- data-sa-034
- data-sa-035
- data-sa-037
- data-sa-041
- data-sa-044
- data-sa-047
- data-sa-065
- data-sa-067
- data-sa-068

## G  LARGE LANGUAGE MODEL USAGE FOR WRITING

In this paper, we employ LLMs—specifically Gemini and ChatGPT—as general-purpose writing tools. Draft text is provided to these models, which are then asked to improve the writing by correcting grammatical errors and refining the structure. The edited text is then verified and edited if needed. The use of LLMs in this paper is limited strictly to text refinement. They were not employed for tasks such as generating any new content or references.

