# OpenReview forum: "LLM-based Multi-Agent Blackboard System for Information Discovery in Data Science"
_ICLR.cc/2026/Conference — Submitted to ICLR 2026_

### Official Review · Reviewer_HNzs · 2025-10-18

**Soundness:** 2
**Presentation:** 2
**Contribution:** 2
**Rating:** 4
**Confidence:** 4

**Summary:**

The paper proposes a multi-agent communication framework for data discovery in data science based on a blackboard architecture. Instead of a master–slave controller assigning subtasks, a main agent posts requests (what information/data is needed) to a shared blackboard. Subordinate agents—each specializing in a partition of the data lake or general retrieval—self-select to respond if they can help. This shifts coordination from centralized routing to decentralized volunteering, improving scalability and flexibility when data lakes contain thousands of heterogeneous files. The system is evaluated on KramaBench and modified DS-Bench/DA-Code with explicit discovery phases, showing 13–57% relative gains in end-to-end task success and up to 9% relative F1 improvement in file discovery over strong baselines (RAG, master–slave multi-agent), across both proprietary and open-source LLMs.

**Strengths:**

Scalable, decentralized coordination: The blackboard lets agents volunteer based on capability, avoiding brittle central task routing and scaling better to large, heterogeneous data lakes.
Practical for real data discovery: Partitioning the lake and binding agents to clusters reduces context overload and noise sensitivity versus single-agent or naïve RAG setups.
Consistent empirical gains and generality: Outperforms RAG and master–slave baselines across multiple benchmarks and LLM types, indicating robustness and portability of the paradigm.

**Weaknesses:**

- Incremental novelty: The “blackboard” paradigm is a classic AI architecture; the paper mostly repackages it for LLM agents without a clear, principled advance in control or learning. The comparison to master–slave is framed conceptually, but the paper lacks a formal analysis of when/why blackboard wins (e.g., under capability uncertainty, overlapping expertise, or partial observability).
- No isolation of the communication factor: what if master–slave uses the same file/search agents but with learned routing or voting?
- Data-lake partitioning is done by filename-only clustering with Gemini. This is fragile for enterprise lakes (cryptic names, mixed schemas) and may bias results. No ablation compares filename-only vs content-aware clustering (embeddings, schema inference), nor sensitivity to cluster granularity.
- The paper repeatedly refers to the benchmark as “DS-Bench,” but the correct name is DSBench

**Questions:**

See weakness

---

> ### Author Response · Authors · 2025-11-21
> **Rebuttal 4-1**
>
> >Incremental novelty: The “blackboard” paradigm is a classic AI architecture; the paper mostly repackages it for LLM agents without a clear, principled advance in control or learning. The comparison to master–slave is framed conceptually, but the paper lacks a formal analysis of when/why blackboard wins (e.g., under capability uncertainty, overlapping expertise, or partial observability).
>
> We respectfully disagree with the assessment that the novelty is incremental. Our core technical contribution is the first principled demonstration of why and when the blackboard architecture is superior to the de facto master-slave model in the new era of high-capability LLM agents. We agree with the reviewer's insightful framing that the central question is when the blackboard wins—specifically, as they note, under conditions of "capability uncertainty, overlapping expertise, or partial observability." We would like to clarify that our paper's experimental design is explicitly built to provide exactly this formal analysis.
>
> The reviewer's concern about "overlapping expertise" is the precise scenario we model. In our setup (detailed in Table 3), we employ an average of 20 distinct "File Agents" alongside one Search Agent. All 20 File Agents possess the same overlapping expertise (the capability to read files), but they have different, siloed knowledge (each agent only sees one data cluster). Additionally, this design directly creates the condition of "partial observability" for any central master. A Master-Slave model fails here because the master agent cannot know which of the 20 File Agents holds the relevant information. Such an example is shown in Figure 18 in Appendix D. It knows what it wants (a file) but not who to ask, creating a critical knowledge bottleneck. The Blackboard model thrives by inverting this. The request is posted to the shared space, and the 20 agents—who have full observability of their own knowledge—can opportunistically self-select and respond.
>
> Therefore, our paper's comparison is not merely "conceptual." Our empirical results serve as the formal analysis the reviewer is asking for. We demonstrate that as soon as you introduce distributed knowledge and overlapping expertise (the reality of complex data systems), the master-slave model's performance collapses while the blackboard's scales. While the blackboard is a classic concept, it was under-explored because 1990s-era agents lacked the reasoning to be autonomous. The "principled advance" we show is that modern LLMs finally fulfill the original vision of the architecture, enabling truly autonomous, reasoning-based coordination.
>
> >No isolation of the communication factor: what if master–slave uses the same file/search agents but with learned routing or voting?
>
> This is a crucial point. We would like to clarify that our Master-Slave baseline already has access to the exact same set of sub-agents as the Blackboard system. We make sure to clarify this in the revised manuscript. The agents are identical; the only difference being tested is the communication paradigm. Thus, our setting already isolated the communication and studied this aspect. The reviewer's suggestion of adding a 'learned router' is insightful, but it's not feasible for this problem domain. The KramaBench benchmark, by design, is a zero-shot setting. There is no training set with ground-truth routing labels, so an agent cannot be trained to know which of the file agents has the correct data. This constraint is preventing us from using this baseline.

---

> > ### Author Response · Authors · 2025-11-21
> > **Rebuttal 4-2**
> >
> > >Data-lake partitioning is done by filename-only clustering with Gemini. This is fragile for enterprise lakes (cryptic names, mixed schemas) and may bias results. No ablation compares filename-only vs content-aware clustering (embeddings, schema inference), nor sensitivity to cluster granularity.
> >
> > We thank the reviewer for this insightful suggestion. The "filename-only" clustering was indeed a simplification, and we agree that a content-aware approach is more robust.
> >
> > To address this, we have conducted the exact ablation study and sensitivity analysis the reviewer proposed. We implemented a new content-aware clustering pipeline by encoding each file’s content using E5-Large embeddings [1] and then applying the KMeans algorithm.
> >
> > For the sensitivity analysis on cluster granularity, we evaluated this new pipeline with K=2, 4, and 8 clusters. We focused this computationally intensive experiment on the larger Legal and Astronomy subsets, which contain at least 100 files each. Our results on content-based clustering, presented in the new Figure 21 in Appendix E, show that performance consistently improves as the number of clusters increases. This confirms that finer-grained clusters allow sub-agents to specialize more effectively on the given data files.
> >
> > Furthermore, Figure 22 in Appendix E directly compares the two clustering strategies. As the reviewer correctly hypothesized, the content-based clustering with E5-Large consistently outperforms our original filename-based approach. We believe this is a strong positive result, as it demonstrates that our Blackboard framework is not reliant on a single, fragile clustering method. In fact, it generalizes effectively and its performance improves when integrated with a more robust, content-aware partitioning scheme. Therefore, regardless of the choice of clustering algorithm to partition data lake, the proposed idea of blackboard system performs well and is generalizable.
> >
> > [1] Liang Wang, undefined., et al, "Text Embeddings by Weakly-Supervised Contrastive Pre-training," 2024.
> >
> > >The paper repeatedly refers to the benchmark as “DS-Bench,” but the correct name is DSBench
> >
> > Thanks for noting this. We made sure that this is fixed in the revised manuscript.

---

### Official Review · Reviewer_f4CA · 2025-10-25

**Soundness:** 2
**Presentation:** 3
**Contribution:** 2
**Rating:** 2
**Confidence:** 3

**Summary:**

This paper introduces a multi-agent system to find relevant information from large data lakes. Instead of using a master-slave model, this paper proposes a multi-agent communication paradigm inspired by the blackboard architecture of traditional AI. Experiments on three data discovery benchmarks show that the proposed method significantly outperforms existing techniques (RAG and master-slave).

**Strengths:**

1.  Developing an LLM-driven system for automating end-to-end data science pipelines is interesting and can augment human analysts.

2. This paper is well-organized and easy to read.

**Weaknesses:**

1. The technical contribution of the work is marginal. The whole pipeline is somewhat not novel and relies on LLM API calls.

2. The proposed framework relies on multiple LLM API calls throughout its pipeline. A comparative analysis of its computational cost and latency against the baseline methods should be included to provide a complete assessment of its efficiency. In particular, we usually use BM25 and BGE embedding to retrieve relevant files from a large pool in practice. The proposed method may face scalability challenges when retrieving relevant files from a large candidate pool, leading to potentially prohibitive latency in real-world applications.

3. More advanced baselines should be compared in terms of both automatic data science and file discovery.

**Questions:**

1. The proposed framework relies on multiple LLM API calls throughout its pipeline. A comparative analysis of its computational cost and latency against the baseline methods should be included to provide a complete assessment of its efficiency. In particular, we usually use BM25 and BGE embedding to retrieve relevant files from a large pool in practice. The proposed method may face scalability challenges when retrieving relevant files from a large candidate pool, leading to potentially prohibitive latency in real-world applications.

2. More advanced baselines should be compared in terms of both automatic data science and file discovery.

---

> ### Author Response · Authors · 2025-11-21
> **Rebuttal 3-1**
>
> >The technical contribution of the work is marginal. The whole pipeline is somewhat not novel and relies on LLM API calls.
>
> We respectfully disagree with the assessment of a marginal contribution. The core technical novelty of our work is not a new model, but the design and validation of the coordination paradigm itself. The reviewer's observation that our pipeline uses LLM calls is, in fact, central to our work's novelty. Classic blackboard systems (like Hearsay-II [1]) were abandoned precisely because the agents of that era lacked the sophisticated reasoning to make the architecture viable. Our work is the first to demonstrate that modern LLMs are powerful enough to finally fulfill the role of the autonomous, opportunistic agents that the blackboard concept originally envisioned. This provides a clear and formal alternative to the de facto standard hierarchical "master-slave" models. We propose a decentralized, non-hierarchical system where agents coordinate opportunistically through a shared blackboard, a fundamentally different approach to communication. This is why the critique of using "LLM API calls" mistakes the implementation for the contribution. The novelty is the protocol, which is model-agnostic. The same architecture would be conceptually unchanged if implemented with fine-tuned models (though we note this is impossible here, as our datasets lack training data). Our contribution is therefore the formal adaptation and evaluation of this non-hierarchical strategy, proving it is a viable and effective alternative to prevailing architectures.
>
> [1] Lee D. Erman, Frederick Hayes-Roth, Victor R. Lesser, and D. Raj Reddy. 1980. The Hearsay-II Speech-Understanding System: Integrating Knowledge to Resolve Uncertainty. ACM Comput. Surv. 12, 2 (June 1980), 213–253. https://doi.org/10.1145/356810.356816
>
> >The proposed framework relies on multiple LLM API calls throughout its pipeline. A comparative analysis of its computational cost and latency against the baseline methods should be included to provide a complete assessment of its efficiency. In particular, we usually use BM25 and BGE embedding to retrieve relevant files from a large pool in practice. The proposed method may face scalability challenges when retrieving relevant files from a large candidate pool, leading to potentially prohibitive latency in real-world applications.
>
> To provide a complete assessment of efficiency and scalability, we conducted a comparative analysis using a random sample of 50 questions from the KramaBench benchmark (combining all tasks). We measured both runtime (latency) and token cost per question.
>
> It is critical to note that in our evaluation, all methods—including the RAG baseline—utilize a ReAct-style reasoning framework. While standard retrieval (BM25/BGE) is indeed fast, it represents a negligible fraction of the total execution time in complex agentic workflows, which are dominated by LLM generation and reasoning loops.
>
> **In terms of Latency:** As shown in Figure 20 in Appendix E, the total runtime for all three systems lies in a narrow band (132.0–145.2 seconds). This parity is achieved because the Blackboard architecture parallelizes sub-agent interactions. Unlike the sequential ReAct loops of the Master–Slave and RAG baselines, our sub-agents process the 'blackboard' state independently and concurrently. Thus, our method does not face the prohibitive latency the reviewer feared, even with a larger candidate pool.
>
> **In Terms of Cost:** We acknowledge that the Blackboard system is more expensive, incurring ~2.3 times the cost of RAG and 1.8 times that of Master–Slave due to increased token usage. However, this investment yields substantial returns: a 54.1% performance gain over RAG and 18.8% over Master–Slave. This demonstrates that the Blackboard architecture offers a favorable trade-off, delivering markedly superior reasoning and answer quality while maintaining a runtime comparable to standard agentic baselines.
>
> Combining retrieval with the blackboard model also offers a practical strategy for reducing costs at scale. For instance, when dealing with data lakes containing millions of files, we can implement an initial retrieval step to isolate the top 1,000 most relevant files, cluster them, and assign a dedicated file agent to oversee each group. This approach significantly lowers the computational overhead by ensuring the agents focus their reasoning on a manageable, high-relevance subset of the data rather than the entire dataset.

---

> > ### Author Response · Authors · 2025-11-21
> > **Rebuttal 3-2**
> >
> > >More advanced baselines should be compared in terms of both automatic data science and file discovery.
> >
> > To further situate our work, we have expanded our comparison to include advanced baselines specifically designed for automatic data science: Data Interpreter [1] and AutoGen [2], both prominent frameworks for multi-agent data analysis. We evaluated these baselines on the KramaBench benchmark, as it is uniquely designed to test both critical aspects of the reviewer's concern: complex data science reasoning and prerequisite file discovery from a large, distributed pool of files. Due to the time constraints of the rebuttal period, we focused our resources on this benchmark as it is the most comprehensive and relevant test for this specific task. The results are presented in the following Table. Our Blackboard system outperforms these strong baselines in five out of six domains, as well as on the overall average.
> > | Method | Archaeology | Astronomy | Biomedical | Environment | Legal | Wildfire | Average |
> > | :--- | :--- | :--- | :--- | :--- | :--- | :--- | :--- |
> > | Data Interpreter | **41.67**% | 12.72% | 28.05% | 9.87% | 30.04% | 59.67% | 30.34% |
> > | AutoGen | 16.67% | 4.39% | 7.25% | 19.38% | 26.38% | 41.76% | 19.30% |
> > | Blackboard | 33.33% | **17.95**% | **36.83**% | **39.31**% | **34.92**% | **62.88**% | **37.53**% |
> >
> > [1] Hong, S., et al, "Data Interpreter: An LLM Agent for Data Science," in Findings of the Association for Computational Linguistics: ACL 2025, 2025, pp. 19796–19821.
> >
> > [2] Qingyun Wu, undefined., et al, "AutoGen: Enabling Next-Gen LLM Applications via Multi-Agent Conversations," in First Conference on Language Modeling, 2024.

---

### Official Review · Reviewer_Fxbd · 2025-10-31

**Soundness:** 2
**Presentation:** 2
**Contribution:** 2
**Rating:** 4
**Confidence:** 4

**Summary:**

The authors propose a multi-agent communication paradigm inspired by the classical blackboard architecture in traditional AI systems. In the proposed framework, a central agent posts queries or requests to a shared blackboard, and a set of autonomous subordinate agents, each responsible for a specific partition of the data lake or a particular retrieval strategy, volunteer to respond based on their capabilities.

**Strengths:**

The paper offers a fresh perspective on coordination and task allocation in multi-agent systems, drawing inspiration from classical AI architectures but applying it to modern data retrieval challenges.

**Weaknesses:**

1. The motivation for adopting a blackboard-style coordination mechanism is somewhat underdeveloped. Given the reasoning ability of LLM-based agents, a master agent could, in theory, dynamically decide which sub-agent to query without needing an explicit shared blackboard. This concern is particularly relevant because the current instantiation only includes two helper agents (file and search agents), making the benefits of the blackboard design less evident.
2. It would make this paper stronger if the authors could show more fine-grained analysis on benchmarks. For example, the performance of different architectures based on the number and types of data files.

**Questions:**

Please check the weaknesses section for more details.

---

> ### Author Response · Authors · 2025-11-21
> **Rebuttal 2**
>
> >The motivation for adopting a blackboard-style coordination mechanism is somewhat underdeveloped. Given the reasoning ability of LLM-based agents, a master agent could, in theory, dynamically decide which sub-agent to query without needing an explicit shared blackboard. This concern is particularly relevant because the current instantiation only includes two helper agents (file and search agents), making the benefits of the blackboard design less evident.
>
> We appreciate the opportunity to clarify the scale of our system, as this directly impacts the architectural choice. While there are only two types of agents (Search and File in this paper but can be easily extended to other types of agents for other tasks), there are many instances of File Agents. As detailed in Table 3 (Appendix A), we deploy one File Agent per data cluster, resulting in an average of 20 agents per dataset. Thus, there are multiple file/search agents to select from.
>
> While a Master Agent could easily decide it needs a file agent (capability selection), it cannot know which specific file agent possesses the relevant data without prior knowledge of all file contents. Additionally, this situation is getting more problematic as the number of file agents grows because the master-slave paradigm needs to ask all agents sequentially to determine who has the knowledge. An example of such a case is shown in Figure 18 in Appendix D. The blackboard architecture is essential here because it allows the sub-agents to self-identify based on their specific knowledge, rather than relying on a central master to route queries to the correct data cluster.
>
> Additionally, Table 1 demonstrates that the blackboard system outperforms the master-slave baseline on end-to-end performance. Furthermore, Table 2 indicates that the master-slave system suffers from lower file discovery performance (F1-measure), providing empirical support for the claims discussed in the previous paragraph.
>
> >It would make this paper stronger if the authors could show more fine-grained analysis on benchmarks. For example, the performance of different architectures based on the number and types of data files.
>
> Thanks for this suggestion. We report relative, rather than absolute, performance gains to ensure fair comparison across datasets with varying task difficulties and score (We use the introduced metric for each task in KramaBench) ranges. Absolute improvements can be misleading when the baseline performance or achievable score range differs substantially. For instance, a 5-point gain might be minor in an easy task with high baseline scores but significant in a challenging one. Relative gain normalizes these differences by measuring proportional improvement with respect to the best baseline (master-slave system), enabling consistent comparison across heterogeneous tasks. Using this, we analyze the scalability of the Blackboard architecture compared to the Master–Slave baseline across datasets with different data lake sizes (Figure 19 in Appendix E). Each point corresponds to a distinct task domain, and a regression line reveals a clear positive correlation between data lake size and relative performance gain. We observed that the Blackboard system has the highest relative gains compared to baselines (70% and 211%) over the master-slave system on the two datasets with the largest number of files (DA-Code with 145 and Astronomy from KramaBench with 1556 data files). This indicates that the Blackboard system scales more effectively as data environments grow larger, whereas the Master–Slave system exhibits limited scalability under such conditions.

---

### Official Review · Reviewer_aVX6 · 2025-11-01

**Soundness:** 3
**Presentation:** 4
**Contribution:** 4
**Rating:** 6
**Confidence:** 4

**Summary:**

This paper proposes a novel multi-agent system for information discovery in data science. Specifically, it proposes a blackboard multi-agent system in which the main agent does not assign tasks to subordinate agents but posts requests to the blackboard. And the subordinate agents autonomously decide whether to respond based on their expertise. The paper evaluates its proposed agent system on three benchmarks including KramaBench, DSBench, and DA-Code and shows its advantage over existing agent systems related to RAG and Master-Slave systems.

**Strengths:**

1. The blackboard multi-agent system is an interesting idea. Similar systems are practically used in large-scale Internet services such as Uber and other on-demand shared riding platforms. It is excellent to apply similar ideas to LLM-based multi-agent systems.

2. The experiments on three benchmarks including KramaBench, DSBench, and DA-Code show advantages over popular agent systems related to RAG and Master-Slave systems.

**Weaknesses:**

1. It would be interesting to further analysis how the “Answering” agents behave in the multi-agent system. For instance, how to handle cases that none of the existing subordinate systems choose to respond? Or, on the opposite, how to handle cases that multiple subordinate systems are competing to respond? More analysis or details would further enhance the depth of the paper.

2. The paper seems to lack the integration or comparison with multi-turn agent system like the ReAct systems. The agent performance typically improves significantly by planning, acting, and reflecting for multiple turns. It is unclear how the blackboard agent system can be integrated into the multi-turn ReAct framework.

3. It is unclear why or how the work modifies DSBench and DA-Code for evaluation. It is recommended to release the modification details for the community to reproduce and align the results. Also, for DSBench, it is unclear whether the data analysis task or data modeling task are considered in the evaluation.

**Questions:**

Typo & Formatting: The Jing et al., 2025 paper uses DSBench instead of DS-Bench. It is recommended to align with the original paper.

---

> ### Author Response · Authors · 2025-11-21
> **Rebuttal 1**
>
> We would like to thank the reviewer for their support of our work. In response to the mentioned weaknesses:
>
> >It would be interesting to further analysis how the “Answering” agents behave in the multi-agent system. For instance, how to handle cases that none of the existing subordinate systems choose to respond? Or, on the opposite, how to handle cases that multiple subordinate systems are competing to respond? More analysis or details would further enhance the depth of the paper.
>
> This is an insightful question. While a full quantitative trace of these dynamic, LLM-driven interactions is complex to model, we have conducted qualitative observations to analyze these exact scenarios.
>
> When no subordinate agent responds: We observed that the “Answering” agent exhibits a retry mechanism. It typically reformulates its request–perhaps simplifying the language or breaking down the query–and re-submits it. If it fails to elicit a response after several attempts, it concludes that the required information is unavailable. Sometimes, it starts searching for the local files itself by generating codes or reports the information does not exist or synthesizes a final answer from any partially gathered information. An example of this is shown in Figure 17 in the revised manuscript in Appendix D.
>
> When multiple subordinate agents compete to respond: The Answering agent is designed to function as an integrator. It receives all contributions and is tasked with combining them. It evaluates the relevance and completeness of each response, selecting and combining the most important information to construct a single, comprehensive answer.
>
> >The paper seems to lack the integration or comparison with multi-turn agent system like the ReAct systems. The agent performance typically improves significantly by planning, acting, and reflecting for multiple turns. It is unclear how the blackboard agent system can be integrated into the multi-turn ReAct framework.
>
> We appreciate the opportunity to clarify this point. As detailed in lines 181-182, the ReAct framework is fundamental to our implementation. Both our proposed method and the baselines (RAG and Master-Slave) are designed such that every agent follows the standard ReAct cycle: they observe the current context, engage in reasoning (utilizing reasoning-focused LLMs), and then execute actions. Therefore, the multi-turn reasoning and acting dynamics described by the reviewer are already intrinsic to our system and baselines.
>
> >It is unclear why or how the work modifies DSBench and DA-Code for evaluation. It is recommended to release the modification details for the community to reproduce and align the results. Also, for DSBench, it is unclear whether the data analysis task or data modeling task are considered in the evaluation.
>
> We provided the steps and details of the filtering process for the dataset in Appendix A and the example ids we used after filtering in Appendix F. Since we do not change the questions or data files, the IDs are enough for reproducing our results. To summarize, we use the analysis task from the DS-Bench.  We exclude questions that request performing a specific operation on a particular column of a data file when the column’s meaning or semantics are insufficiently described. In such cases, it would be infeasible to accurately identify the target column within the data lake, given that multiple files may contain columns with the same name. Furthermore, questions that focus solely on the operation itself–assuming access to only a single file–are also excluded, as they lack sufficient contextual information for meaningful retrieval or reasoning about the data file that needs to be discovered from the data lake. Finally, since our goal is to study information discovery in data science, we also exclude questions that can be answered without accessing any data files, as these are general data science questions that the model has the knowledge about them in its parameters and does not need any data files. These filtering steps have been done manually by humans.
>
> About typo: thanks for pointing this out, we made sure that this is fixed in the paper (in the revised uploaded version).

---

> > ### Comment · Reviewer_aVX6 · 2025-11-24
> > **On rebuttal**
> >
> > Thanks for the detailed response. I will put 6 as my overall rating after rebuttal.

---

### Author Response · Authors · 2025-11-30
**Summarizing Responses to reviwers for AC**

Dear Area Chair,

We thank the reviewers for their constructive feedback. In response, we have revised our manuscript and conducted **four new sets of experiments** and **two more case studies of the agents behaviors** to address concerns regarding scalability, cost, advanced baselines, and implementation robustness. We believe these additions provide strong empirical evidence supporting the superiority of the Blackboard architecture over Master-Slave and RAG paradigms in data science contexts.

Below is a summary of the key changes and how they address specific reviewer concerns:

### 1. Comparison with SOTA Agentic Frameworks (Addressing Reviewer f4CA)
Reviewer f4CA suggested comparing our method against more advanced data science agents.
* **New Experiment:** We compared the Blackboard System against **Data Interpreter** (Hong et al., 2025) and **AutoGen** (Wu et al., 2023) on the KramaBench benchmark.
* **Result:** As shown in **Table 4 (Appendix E)**, our method outperforms these competitive baselines in 5 out of 6 domains and achieves a higher overall average (37.53% vs 30.34% and 19.30%), demonstrating superior coordination capabilities.

### 2. Cost and Latency Analysis (Addressing Reviewer f4CA)
Reviewer f4CA raised concerns regarding the computational cost and latency of the system compared to standard retrieval methods.
* **New Analysis:** We conducted a runtime and cost analysis on 50 random samples from KramaBench.
* **Result:** As shown in **Figure 20 (Appendix E)**, while our method incurs higher token costs (~2.3x RAG), the **latency is comparable** to baselines (139.9s vs 145.2s for Master-Slave) due to the parallel nature of sub-agent processing. Given the substantial performance gain (+54% relative to RAG), we argue this is a favorable trade-off for complex reasoning tasks.

### 3. Scalability and "Why Blackboard Wins" (Addressing Reviewers Fxbd & HNzs)
Reviewers Fxbd and HNzs asked for a formal analysis of when the Blackboard architecture outperforms Master-Slave, specifically regarding data lake size and routing.
* **New Analysis:** We analyzed the relative performance gain of Blackboard over Master-Slave as a function of data lake size.
* **Result:** **Figure 19 (Appendix E)** shows a clear positive correlation: as the data lake grows (increasing partial observability and routing difficulty for a central master), the Blackboard system's advantage widens significantly (up to 211% relative gain on the largest dataset).
* **Qualitative Evidence:** We added **Figure 18 (Appendix D)** to illustrate a specific failure mode where a Master-Slave controller fails to route queries to the correct siloed agent due to lack of global knowledge—a bottleneck the decentralized Blackboard design eliminates.

### 4. Robustness of Clustering Strategy (Addressing Reviewer HNzs)
Reviewer HNzs noted that filename-based clustering might be fragile and suggested content-aware clustering.
* **New Ablation Study:** We implemented a content-based clustering pipeline using **E5-Large embeddings and KMeans**, testing sensitivity to cluster granularity (K=2, 4, 8).
* **Result:** **Figures 21 & 22 (Appendix E)** demonstrate that content-based clustering consistently outperforms filename-based clustering and that performance improves with finer-grained clusters. This confirms our framework generalizes well to robust, embedding-based partitioning schemes.

### 5. Agent Behavior & Reproducibility (Addressing Reviewer aVX6)
* **Clarification:** We added **Figure 17 (Appendix D)** to visualize the "retry and synthesis" mechanism the Main Agent employs when no sub-agents volunteer.
* **Reproducibility:** We have added **Appendix F**, listing the specific Example IDs used for the filtered DSBench and DA-Code datasets to ensure exact reproducibility of our results.

More detailed response and discussion regarding each concern is directly replied to the reviewers (please refer to each review specific response). We believe these comprehensive updates resolve the reviewers' concerns regarding novelty, efficiency, and robustness, confirming the Blackboard system as a scalable and effective paradigm for modern LLM-based multi-agent systems. We kindly ask the AC to take these updates into consideration, as they address the feedback directly and could have positively influenced the reviewers’ scores had the conversation continued.


Best regards,
The Authors

---

### Meta-Review · Area_Chair_8zoM · 2025-12-20

**Summary:**

**Paper summary.** This paper proposes a multi-agent “blackboard” architecture for information discovery in data science settings with large data lakes. A main agent posts requests to a shared board, and many sub-agents (e.g., file agents bound to data clusters, plus search agents) decide whether to respond based on their expertise and local context. The paper positions this as an alternative to a master–slave controller and reports gains on multiple benchmarks.

**What happened in the discussion.** The reviews were mixed (2/4/4/6) and the debate was mainly about novelty and evidence. Reviewers asked: (1) what is really new beyond applying a classic blackboard idea to LLM agents, (2) whether the benefits are clear given the presented agent types, (3) how the system behaves in edge cases (no agent responds / too many respond), (4) whether the clustering/partitioning method is fragile, and (5) whether cost/latency is competitive given many LLM calls (and whether stronger multi-agent baselines should be included). In rebuttal, the authors responded with several concrete additions: they clarified that while there are only a few agent *types*, there are many file-agent instances (roughly one per cluster), they added qualitative behavior analysis for “no response” and “too many responders”, they added new baseline comparisons on KRAMABENCH (including Data Interpreter and AutoGen), and they ran an ablation replacing filename-based clustering with content-aware clustering using E5 embeddings and varying cluster counts.

**My assessment as AC.** The rebuttal improved the submission and addressed some technical concerns (especially clustering ablation and stronger baselines). Still, the core contribution remains a systems design that is conceptually straightforward, and it is hard to separate what drives the gains (blackboard vs. partitioning vs. prompt/agent engineering). In this batch, I am not convinced the novelty and evidence are strong enough compared to the top submissions. I also think a clearer, quantitative cost/latency analysis (token/call budget and runtime) would be needed to fully justify the approach.

**Decision.** Reject. This is not saying the paper is not useful; the idea can work in practice. The rejection is mainly due to competitiveness and the fact that different people weigh “new architecture” vs. “strong evidence + cost analysis” differently. I encourage the authors to (a) add a clean ablation that isolates the blackboard mechanism from other components, and (b) include a clear cost/latency breakdown vs. strong baselines. With those changes, the paper could be much more competitive at a future venue.

**Reviewer Concerns:**

- **Reviewer HNzs (rating 4, confidence 4)**: Liked the framing but questioned novelty and asked for stronger isolation of what makes blackboard win and how it behaves vs. alternatives. Authors provided more explanation but isolation is still limited. **Status:** partially resolved.
- **Reviewer f4CA (rating 2, confidence 3)**: Called the technical contribution marginal and asked for cost/latency analysis and stronger baselines. Authors added stronger baselines (Data Interpreter, AutoGen) on KRAMABENCH, but cost/latency remains under-reported. **Status:** partially resolved.
- **Reviewer Fxbd (rating 4, confidence 4)**: Questioned whether a blackboard is needed given LLM reasoning and noted the presented “two helper agent types” might not demonstrate the benefit; authors clarified there are many file-agent instances and explained scalability. **Status:** mostly resolved.
- **Reviewer aVX6 (rating 6, confidence 4)**: Asked for analysis of answering-agent behavior in “no responder / multiple responders” settings and for broader comparison; authors added qualitative analysis and details. Reviewer later commented they would set rating to 6 after rebuttal. **Status:** mostly resolved.

**Reviewer Scores:**

- **aVX6 (6,4)**: Explicitly confirmed rating 6 after rebuttal.
- **HNzs (4,4)**: Likely unchanged.
- **Fxbd (4,4)**: Likely unchanged.
- **f4CA (2,3)**: Could move up slightly due to added baselines, but likely remains negative without clear cost reporting.

---

### Decision · Program_Chairs · 2026-01-26

Reject